# Do LLMs estimate uncertainty well in instruction-following?

**Juyeon Heo[1],[†]  Miao Xiong[3],[†]  Christina Heinze-Deml[2] Jaya Narain[2]**
[1]University of Cambridge    [2]Apple    [3]National University of Singapore
jh2324@cam.ac.uk   jnarain@apple.com

## Abstract

Large language models (LLMs) could be valuable personal AI agents across various domains, provided they can precisely follow user instructions. However, recent studies have shown significant limitations in LLMs' instruction-following capabilities, raising concerns about their reliability in high-stakes applications. Accurately estimating LLMs' uncertainty in adhering to instructions is critical to mitigating deployment risks. We present, to our knowledge, the first systematic evaluation of uncertainty estimation abilities of LLMs in the context of instruction-following. Our study identifies key challenges with existing instruction-following benchmarks, where multiple factors are entangled with uncertainty stemming from instruction-following, complicating the isolation and comparison across methods and models. To address these issues, we introduce a controlled evaluation setup with two benchmark versions of data, enabling comprehensive comparison of uncertainty estimation methods under various conditions. Our findings show that existing uncertainty methods struggle, particularly when models make subtle errors in instruction following. While internal model states provide some improvement, they remain inadequate in more complex scenarios. The insights from our controlled evaluation setups provide crucial understanding of LLMs' limitations and potential for uncertainty estimation in instruction-following tasks, paving the way for more trustworthy AI agents.[1]

## 1 Introduction

Large language models (LLMs) have garnered interest for their potential as personal AI agents across various domains, such as healthcare, fitness, nutrition, and psychological counseling (Li et al., 2024; Wang et al., 2023a; Tu et al., 2024). A key to building safe and useful personal AI agents with LLMs lies in their ability to follow instructions precisely. Deployed models must adhere to the constraints and guidelines provided by users to ensure that the outputs are both aligned with user intentions and safe. Yet recent research has exposed significant limitations in LLMs' ability to follow instructions (Zhou et al., 2023; Zeng et al., 2023; Qin et al., 2024; Xia et al., 2024; Kim et al., 2024; Yan et al., 2024). For example, even large models like GPT-4 achieve only around 80% instruction-following accuracy on simple and non-ambiguous instructions from benchmark datasets, and smaller models perform even worse, with accuracy less than 50% (Sun et al., 2024).

Since LLMs are prone to errors, their ability to accurately assess and communicate their own uncertainty is essential. This becomes particularly important in high-stakes applications, where mistakes can have serious consequences. For instance, an LLM developed for personal psychological counseling must strictly adhere to guidelines that avoid topics that might potentially cause trauma. If the LLM misinterprets or deviates from these instructions but accurately recognizes and signals high uncertainty, it could prompt further review or intervention, thereby preventing the delivery of potentially harmful advice.

However, uncertainty estimation in instruction-following tasks has received limited attention, with most research focusing on fact-based tasks like question answering and summarization (Fadeeva

---

[†] Work done while at Apple.
[1]Code and data are available at https://github.com/apple/ml-uncertainty-llms-instruction-following

et al., 2023; Kuhn et al., 2023; Xiong et al., 2023; Ye et al., 2024), where factual correctness is the primary concern. In contrast, as shown in Figure 1, instruction-following tasks focus on whether a model's response adheres to a set of given instructions, rather than estimating the factual accuracy. Given these different source of uncertainty, it is unclear whether existing methods, which are typically designed for estimating factual uncertainty, can accurately capture uncertainty in instruction following. For example, while semantic entropy (Farquhar et al., 2024) is considered the gold standard for fact-based tasks, it may not be suitable for instruction-following. Both responses, 'Regular exercise strengthens muscles' and 'Regular exercise reduces stress', follow the given instruction in Figure 1 but convey different semantic meanings, leading to a high semantic uncertainty score, which inaccurately reflects instruction-following performance. This example highlights the need for frameworks tailored to evaluating methods and models for uncertainty estimation in instruction-following tasks.

To evaluate how well existing uncertainty estimation methods and models perform on instruction-following tasks, we evaluate six uncertainty estimation methods across four LLMs on the IFEval benchmark dataset (Zhou et al., 2023). However, we find that multiple factors are entangled in existing benchmarks. For instance, uncertainty can stem from both task execution quality and instruction following, making it difficult to isolate and directly compare methods and models based solely on their ability to estimate instruction-following uncertainty. To address these issues, we design a new benchmark dataset with two versions to enable a more controllable and fine-grained evaluation. The **Controlled** version provides a structured assessment by removing confounding factors and offering tasks with varying difficulty levels, split into Controlled-Easy, where correct and incorrect responses are easy to distinguish, and Controlled-Hard, which focuses on more subtle errors. In contrast, the **Realistic** version uses naturally generated LLM responses, retaining real-world signals. Together, these datasets provide a comprehensive framework for evaluating uncertainty estimation methods and models under both controlled and real-world conditions.

Our analysis revealed several key findings from the controlled evaluation: 1) Verbalized method consistently outperforms logit-based methods like perplexity in the Controlled-Easy setting, where correct and incorrect responses are relatively easier to distinguish. Specifically, normalized p(true) (Kadavath et al., 2022) proves to be a reliable uncertainty method across both Controlled-Easy and Realistic settings. 2) Smaller models often outperform larger ones in verbalized confidence, suggesting that factors beyond model size, such as tuning or architecture, may contribute to better uncertainty estimation in certain tasks. 3) Probes relying on the internal states of LLMs outperform logit-based and verbalized confidence, highlighting promising directions for future work. 4) In more challenging tasks like Controlled-Hard, which involve subtle off-target responses, all approaches including internal representations struggle to estimate uncertainty accurately, pointing to inherent limitations in LLMs' ability to handle complex uncertainty. These findings from our controlled evaluation setups provide crucial insights into the limitations and potential of LLMs for uncertainty estimation in instruction-following tasks, towards more trustworthy AI agents.

## 1.1 CONTRIBUTIONS

- **Systematic Evaluation:** We present the first systematic evaluation of uncertainty estimation methods in instruction-following tasks, addressing a gap in existing research.

- **Benchmark Dataset:** We identify key challenges in existing datasets and introduce a new benchmark dataset specifically tailored for direct comparison and fine-grained analysis of uncertainty estimation methods and models in both controlled and real-world conditions.

- **Findings:** Our evaluation results highlight the potential of self-evaluation and probing methods and point out limitations in handling more complex tasks, underscoring the need for further research to advance uncertainty estimation in instruction-following tasks.

## 2 UNCERTAINTY ESTIMATION ABILITY IN INSTRUCTION-FOLLOWING ON IFEVAL

In this section, we evaluate LLMs' uncertainty estimation abilities using the IFEval dataset (Zhou et al., 2023), applying six baseline methods across four LLMs. We selected IFEval because it is designed so that a simple and deterministic program can verify whether a response follows the in-

Figure 1: **Why evaluating uncertainty estimation ability in instruction-following matters**. Uncertainty in instruction-following distinct from factual correctness, as illustrated by this example. While both responses shown are factually correct, the first fails to follow the instruction, resulting in high uncertainty from instruction-following despite low uncertainty from factuality. The second response adheres to the instruction, with low uncertainty in both areas. Prior work on uncertainty has focused primarily on factual correctness, underscoring the need for an evaluation framework for instruction-following tasks.

structions. This enables a fully automatic and accurate assessment of a model's instruction-following capability, thereby minimizing uncertainties from ambiguous evaluation criteria.

## 2.1 METHODS

**Data** We evaluate uncertainty estimation with the IFEval dataset (Zhou et al., 2023), which is designed to evaluate the instruction-following ability of LLMs on 25 verifiable instruction types under 9 categories across 541 total prompts. Each prompt consists of two components: a task and an instruction, where the *instruction* specifies the action (e.g., "please do not use keywords", "please start/finish your response with exact sentence") and the *task* provides the context for executing the instruction (e.g., "please write a resume", "please give a summary about solar system").

**Models and Metrics** We evaluate four LLMs of varying sizes: LLaMA2-chat-7B (Touvron et al., 2023), LLaMA2-chat-13B (Touvron et al., 2023), Mistral-7B-Instruct-v0.3 (Jiang et al., 2023), and Phi-3-mini-128k-instruct (Abdin et al., 2024). To avoid randomness in decoding, we employ greedy decoding without sampling. Area Under the Receiver Operating Characteristic curve (AUROC) (Pedregosa et al., 2011) is used to measure if the models' uncertainty estimation matches the ground truth labels on correctness in instruction following, generated using the automated evaluation functions from IFEval.

**Baseline uncertainty estimation methods** To evaluate uncertainty in instruction-following, we employ several baseline methods, including self-evaluation of their own uncertainty (verbalized confidence, normalized p(true) and p(true)), logits-based method (perplexity, sequence probability, and mean token entropy), and probing method:

- **Verbalized confidence** (Lin et al., 2022; Xiong et al., 2023; Tian et al., 2023): The model's self-reported confidence, scored from 0 to 9, indicating its perceived likelihood that the response correctly follows instructions. Detailed prompts are in the Appendix A.2.

- **Normalized p(true) and p(true)** (Kadavath et al., 2022; Lin et al., 2022): These methods assess the probability of the 'true' token, calculated from a binary choice prompt. We modified the calculation to determine the probability of token 'A' given the prompt: "*Does the response: (A) Follow instructions (B) Not follow instructions. The answer is:* ". Normalized p(true) adjusts for biases by considering both tokens of 'true' and 'false' probabilities: $p(\text{true})/(p(\text{true}) + p(\text{false}))$, here is $p(\text{A})/(p(\text{A}) + p(\text{B}))$.

- **Perplexity and Sequence probability** (Jelinek et al., 1977; Fomicheva et al., 2020): Perplexity measures the likelihood of generating a given sequence: $\exp\left\{-\frac{1}{t}\sum_{i=1}^{t}\log p_\theta(x_i \mid x_{<i})\right\}$ where $t$ is the sequence length. Sequence probability, an unnormalized version, is calculated as: $\exp\left\{-\sum_{i=1}^{t}\log p_\theta(x_i \mid x_{<i})\right\}$ , making it more sensitive to the length of the response, with longer sequences generally having lower probabilities.

| Model | AUC of uncertainty methods | | | | | | SR |
|---|---|---|---|---|---|---|---|
| | Verbal | Perplexity | Sequence | Nor-p(true) | p(true) | Entropy | |
| LLaMA2-chat-13B | 0.53 | 0.44 | 0.61 | 0.47 | 0.51 | 0.48 | 0.57 |
| LLaMA2-chat-7B | 0.53 | 0.43 | 0.54 | 0.52 | 0.51 | 0.44 | 0.59 |
| Mistral-7B-Instruct-v0.3 | 0.50 | 0.48 | 0.56 | 0.57 | 0.47 | 0.50 | 0.64 |
| Phi-3-mini-128k-instruct | 0.53 | 0.43 | 0.48 | 0.55 | 0.45 | 0.45 | 0.54 |

Table 1: **AUROC for baseline uncertainty estimation methods applied to the IFEval dataset** (Zhou et al., 2023). AUROC measures how well each method's uncertainty estimates align with the ground truth regarding correct or incorrect instruction-following across four LLMs. Success Rate (SR) represents the model's instruction-following accuracy, calculated using the IFEval evaluation function. Notably, LLMs struggle to estimate uncertainty in instruction-following tasks hover around chance levels (between 0.43 and 0.53).

- **Mean token entropy for LLMs** (Fomicheva et al., 2020): Entropy measures uncertainty based on token prediction variability: $H = -\frac{1}{t} \sum_{i=1}^{t} p_\theta(x_i \mid x_{<i}) \log p_\theta(x_i \mid x_{<i})$

- **Probing**: Drawing inspiration from Liu et al. (2024), which suggests that model representations contain valuable information for uncertainty estimation in tasks like question and answering, we investigated whether this internal knowledge could similarly improve uncertainty estimation in instruction-following tasks. We trained a linear model as an uncertainty estimation function that maps the internal representations of LLMs to instruction-following success labels, where the probability predicted by the linear model used as uncertainty scores. We use layers 16, 32, and 40 and for LLaMA-2-13B-chat and 14, 26, and 32 for other three models. The ground truth for success or failure is determined by a deterministic program that verifies instruction adherence using the IFEval dataset. We train a linear model on representations on instruction-following success labels, optimized with AdamW, a 0.001 learning rate, 0.1 weight decay. The model is trained for 1000 epochs on 70% training set and is evaluated on 30% test set. We refer to this as **Probe**.

## 2.2 FINDINGS

Table 1 summarizes uncertainty evaluation results using IFEval.

**LLMs struggle to estimate uncertainty in instruction-following.** Average AUROC values across models and uncertainty estimation methods hover around chance levels (between 0.43 and 0.53), indicating that the models consistently fail to reliably assess their own uncertainty in instruction-following. This underscores the challenge LLMs face in detecting when their responses deviate from the instructions.

**Sequence probability outperforms perplexity, revealing a length signal in uncertainty estimation.** Sequence probability consistently achieves higher AUROC scores than perplexity across most models. Sequency probability is tied to by sequence length, whereas perplexity is not. For example, in LLaMA2-chat-13B, the AUROC for sequence probability averages 0.61 (above chance) across instruction types, whereas perplexity lags at 0.44. This finding implies that response length may inadvertently provide a signal in some uncertainty estimation metrics, even though it may not correlate directly with the correctness of the response in instruction-following.

**No model or method consistently excels across instruction types.** As shown in Table 5 in Appendix, there is no consistent pattern of performance of uncertainty estimation method or model across different instruction types. This lack of consistency indicates that none of the uncertainty estimation methods evaluated reliably capture uncertainty across all instruction types and models.

## 2.3 CHALLENGES IN EVALUATING UNCERTAINTY ESTIMATION USING EXISTING DATASETS

An instruction-following dataset with clear evaluation criteria, like IFEval, is important for evaluating instruction-following. However, we identified the importance of several additional factors to aid in comparatively evaluating instruction-following *uncertainty* of LLMs. In our systematic evaluation, we identified multiple factors affecting uncertainty that are entangled within naturally generated responses, making it difficult to isolate the strengths and weaknesses of each method or

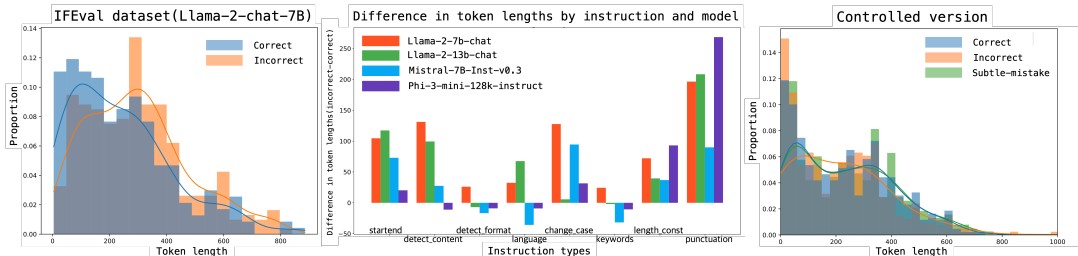

(a) Length distribution on IFEval (b) Length differences by instruction types  (c) Length in Controlled version

Figure 2: **Existing instruction-following datasets only evaluate uncertainty estimation methods in length-biased settings, missing comparisons on controlled, length-neutral conditions.** The distributions are normalized by the total number of responses in each class. (a) Token lengths distribution for LLaMA-2-chat-7B shows that incorrect responses tend to be longer than correct ones. This length signal is prevalent in existing dataset like IFEval, where naturally generated responses are used. (b) Token length differences broken down by instruction type and model, with positive values showing that incorrect responses tend to be longer. However, this trend is not consistent across all instructions or models, highlighting the need for controlled evaluation setups. (c) Token length distribution in the Controlled version of our benchmark, where token length is balanced across correct, incorrect, and subtly incorrect responses. This setup allows us to evaluate uncertainty estimation in both length-biased and length-neutral settings.

model. Below, we outline the primary challenges we identified and present analyses demonstrating each challenge.

**1) Uncertainty estimation methods and models are only evaluated in length-biased settings in existing instruction-following datasets, missing comparisons on controlled, length-neutral conditions.** We observe that token length significantly impacts uncertainty estimation in instruction-following tasks in existing datasets, where responses are not controlled but are generated as part of the evaluation. With datasets like IFEval, naturally generated incorrect responses tend to be longer than correct ones across models. Figure 2a and Appendix Figure 10 present a histogram of token lengths for both response types on four LLMs, while Appendix Table 11 in the provides detailed statistics on response length, including the mean and standard deviation for correct and incorrect responses. Consistently, our analysis show that *incorrect responses tend to be longer than correct ones across different LLMs in IFEval dataset*.

If this pattern holds across instruction types, length could arguably be a reliable signal in evaluating instruction-following uncertainty. However, we found that the relationship between response length and correctness is not uniform (Figure 2b). In instruction types like 'startend' and 'punctuation', incorrect responses were uniformly longer across all models. Conversely, in instruction types such as 'detectable-content', 'keywords', and 'language', correct responses were often longer. These varying patterns are not surprising, as response length is related to what the instruction requires. For instance, an instruction like "please elaborate" would naturally result in a longer response, whereas "make it concise" would lead to a shorter one.

These inconsistencies suggest that length is not a reliable or generalizable signal for uncertainty estimation across all instruction types and models. This highlights the need for a controlled evaluation framework that includes a set of length-neutralized responses. By comparing LLMs and uncertainty estimation methods in both length-biased and length-neutral settings, we can more accurately assess their true performance, independent of confounding factors like token length.

**2) Uncertainty sourced from task execution quality is entangled with uncertainty stemming from instruction-following, complicating accurate evaluation.** In the IFEval data, each prompt consists of two parts: the *task* context (e.g., "Write a brief summary about the solar system") and the specific *instruction* (e.g., "Please do not mention any planet names"). When measuring uncertainty with baseline methods, uncertainty can stem from both task execution quality (i.e., how well the task itself is accomplished – clear, detailed, and informative) and instruction-following accuracy (i.e., whether the instruction is followed). This creates an entanglement that complicates evaluation. For example, consider the response "Objects in space around a star", which is vague and unclear with

| Case | Task quality entanglement | | | Model | Instruction-following eval | | |
|------|---------------|--------------|----------------------|-------|---------|-------|------|
| | Inst-following | Task quality | Verbalized confidence | | Correct | Wrong | Diff |
| Case 1 | o | H | $7.53 \pm 0.49$ | Llama-2-chat-13B | $7.48 \pm 2.78$ | $6.37 \pm 3.40$ | 1.11 |
| Case 2 | o | L | $7.00 \pm 0.63$ | Llama-2-chat-7B | $7.10 \pm 3.24$ | $6.56 \pm 3.15$ | 0.54 |
| Case 3 | x | H | $7.42 \pm 0.51$ | Mistral-7B-Inst-v0.3 | $7.54 \pm 2.95$ | $6.77 \pm 3.29$ | 0.77 |
| Case 4 | x | L | $6.64 \pm 0.33$ | Phi-3-mini-128k-instruct | $6.10 \pm 3.61$ | $5.12 \pm 3.77$ | 0.98 |

Table 2: **Left** Verbalized confidence from the LLaMA-2-chat-7B model on four cases of instruction-following and task quality. GPT-4 evaluates task quality(0-9), where H and L represent high and low task quality(threshold 8), respectively, and o and x indicate whether the responses successfully follow the instructions. It shows that verbalized confidence is affected by task quality, revealing the entanglement between uncertainty from task execution quality and instruction-following. **Right** GPT-4 evaluates instruction-following scores for correct and incorrect responses across four LLMs. The Diff represents the gap between correct and incorrect cases, showing different levels of difficulty in distinguishing between correct and incorrect responses across models.

low task quality but adheres to the instruction of avoiding mentioning planet names. Alternatively, "The solar system consists of a central star surrounded by various celestial bodies, including Earth and Mars" is informative with high task quality but fails to follow the instruction. We observe that task execution quality also influences the models' uncertainty scores. For example, in Table 2, the LLaMA-2-chat-7B model assigns an average verbalized confidence score of 7.0 to the first response grouping (low task quality but correct instruction-following) and 7.4 to the second (high task quality but incorrect instruction-following).

This shows that task quality can confound uncertainty estimation in instruction-following. When task execution quality is not controlled, the uncertainty arising from task completion can over-shadow the uncertainty associated with following instructions, leading to inaccurate evaluations of the model's true instruction-following uncertainty. A controlled setup, where task quality is held constant, can help disentangle these two sources of uncertainty, allowing for a more focused evaluation of models' uncertainty estimation specific to instruction-following. Detailed prompt for evaluating task execution quality can be found in Appendix A.3.

**3) Differences in the severity of instruction-following mistakes across models create inconsistent difficulty levels, complicating model comparisons.** Our primary objective is to assess LLMs' uncertainty estimation capabilities, independent of their instruction-following accuracy in generating responses. However, when using the IFEval dataset, these two factors are entangled, making it difficult to isolate uncertainty estimation from the model's overall instruction-following performance. For example, LLaMA-2-chat-13B, which generally has a higher instruction-following accuracy, tends to make more obvious errors when it fails to follow instructions. On the other hand, LLaMA-2-chat-7B not only makes these obvious mistakes but also exhibits more subtle instruction-following errors, where responses partially follow the instructions but miss specific details. As a result, uncertainty estimation becomes more challenging for LLaMA-2-chat-7B, where it has to measure uncertainty in more nuanced instruction violations, compared to LLaMA-2-chat-13B.

To quantify the difference in task difficulty, we use GPT-4 to score the responses on a scale from 0 to 9 based on their adherence to instructions. Table 2 shows that the score gap between correct and incorrect responses is smaller for LLaMA-2-chat-7B compared to LLaMA-2-chat-13B, highlighting the more subtle nature of 7B's errors. In contrast, 13B's errors are more drastic, making them easier to identify and be recognized as having high uncertainty. This variation in task difficulty across models complicates direct comparisons of their uncertainty estimation abilities. To ensure fair comparisons across models, it is necessary to evaluate models under controlled difficulty levels.

## 3 UNCERTAINTY ESTIMATION ABILITY IN INSTRUCTION-FOLLOWING ON OUR BENCHMARK DATA

The challenges identified in the previous section—length bias, the entanglement of task execution quality with instruction-following, and varying difficulty levels across models—underscore the need for a controlled and robust framework for evaluating uncertainty estimation. To address these issues, we develop a new benchmark dataset comprising two versions: Controlled and Realistic version.

| Case | Example |
|---|---|
| Correct | "Experienced Refinery Operator with 5 years in the chemical industry. Skilled in overseeing refinery operations, maintaining safety protocols, and optimizing processes. Collaborated with **friends** at **Hanson** Chemicals to enhance efficiency. Proven track record of improving production quality." |
| Incorrect | "Professional Refinery Operator with 10 years in the chemical industry. Expert in managing refinery operations. Demonstrated history of elevating production quality." |
| Subtle Error | "Experienced Refinery Operator with 5 years in the chemical industry. Proficient in managing refinery operations, ensuring safety protocols, and enhancing processes. Worked closely with colleagues at **Hanson** Chemicals to boost efficiency. Demonstrated history of elevating production quality." |

Table 3: **Examples of three response types in the Controlled version of our benchmark dataset**. Illustrate responses to the prompt: "Write a short resume for a refinery operator with 5 years of experience in the chemical industry. Include the keywords friends and Hanson". The examples show a Correct response that fully follows instructions, an Incorrect response that omits the keywords, and a Subtle Error that includes only one keyword, highlighting varying degrees of instruction-following.

These versions allow for the evaluation of uncertainty estimation under controlled conditions (Controlled) and real-world scenarios (Realistic).

## 3.1 BENCHMARK DATASET FOR CONTROLLED EVALUATION SETUPS

To disentangle the complexities that can obscure uncertainty estimation, we design two distinct versions of the dataset: **Controlled** and **Realistic**. The Controlled version neutralizes the influence of token length. Meanwhile, the Realistic version leverages actual LLM-generated responses that naturally incorporate real-world signals, including length signal, without manual intervention.

In both versions, we use GPT-4 to filter out low-quality responses, ensuring that the uncertainty being measured comes from instruction-following, not poor task execution (addressing the second challenge in section 2.3). We apply a filtering process using GPT-4 evaluations of task quality of each response on a scale 0-9. Only responses that received a high task quality score ($>8$) are included. Also, these datasets enable using the same responses with all models in the uncertainty evaluation task, controlling the difficulty of uncertainty evaluation across models. This allowing for direct comparisons in uncertainty estimation (addressing the third challenge in section 2.3).

### 3.1.1 CONTROLLED VERSION

In this version, we eliminate the length effect, along with ensuring consistent levels of difficulty across responses to ensure that the evaluation focuses purely on uncertainty estimation. To neutralize the impact of token length, we use GPT-4 to generate both correct and incorrect responses with similar token length (see Appendix for the prompt used to generate responses). Figure 2c shows the absence of length bias in the token length distribution. We introduce two levels of controlled difficulty–**Controlled-Easy** and **Controlled-Hard**–by generating three categories of responses: *completely incorrect, correct, and subtly off-target*. In the Controlled-Easy, we calculate AUROC based on distinguishing between correct and completely incorrect responses, while in the Controlled-Hard, we calculate AUROC based on distinguishing between correct and subtly off-target responses. These are more challenging cases where the responses only slightly deviate from the instructions, testing the model's ability to recognize subtle mistakes. While these mistakes are subtle, they could still be important in real-world deployments. Table 3 shows an example from this version. Additional statistics are provided in Table 7 in the Appendix.

### 3.1.2 REALISTIC VERSION

In the Realistic version, we retain the natural length and signals inherent in responses generated by multiple LLMs (LLaMA2-chat-7B, LLaMA2-chat-13B, Mistral-7B-Instruct-v0.3, Phi-3-mini-128k, and LLaMA2-chat-70B). Here, the goal is to evaluate uncertainty estimation methods in a scenario that reflects actual model behavior. In this version, we do not control for token length, allowing for the natural variance found in actual model-generated responses. Though, we still control for task execution quality and provide generated responses to enable clear comparisons between models.

| Model | Controlled-Easy | | | | | | | Controlled-Hard | | | | | | | Realistic | | | | | | |
|---|---|---|---|---|---|---|---|---|---|---|---|---|---|---|---|---|---|---|---|---|---|
| | Verb | Ppl | Seq | N-p(t) | p(t) | Ent | Probe | Verb | Ppl | Seq | N-p(t) | p(t) | Ent | Probe | Verb | Ppl | Seq | N-p(t) | p(t) | Ent | Probe |
| LLaMa2-13B | 0.67 | 0.62 | 0.48 | 0.61 | 0.46 | 0.57 | **0.79** | 0.52 | **0.60** | 0.52 | 0.54 | 0.44 | 0.57 | 0.58 | 0.51 | 0.52 | 0.61 | 0.53 | 0.48 | 0.46 | **0.64** |
| LLaMa2-7B | 0.64 | 0.61 | 0.48 | 0.54 | 0.45 | 0.54 | **0.72** | 0.53 | **0.57** | 0.51 | 0.51 | 0.45 | 0.53 | 0.51 | 0.48 | 0.51 | **0.62** | 0.53 | 0.47 | 0.50 | 0.61 |
| Mistral-7B | 0.71 | 0.59 | 0.44 | 0.66 | 0.48 | 0.43 | **0.75** | 0.56 | 0.55 | 0.49 | 0.56 | 0.48 | 0.46 | 0.56 | 0.53 | 0.46 | 0.63 | 0.51 | 0.51 | 0.49 | **0.66** |
| Phi-3-mini | 0.64 | 0.58 | 0.42 | 0.72 | 0.56 | 0.55 | **0.79** | 0.55 | 0.53 | 0.47 | **0.62** | 0.48 | 0.52 | 0.53 | 0.49 | 0.42 | 0.63 | 0.56 | 0.51 | 0.51 | **0.72** |

Table 4: **Average AUC across instruction types** for different LLMs and uncertainty estimation methods in three settings. Different uncertainty estimation methods includes Verbalized confidence (Verb), Perplexity (Ppl), Sequence probability (Seq), Normalized p(true) (N-p(t)), p(true), Entropy (Ent), and linear probing on internal states (Probe). Bold values indicate the best-performing method for each model and condition, while underlined values denote the second-best performing method.

## 3.2 RESULTS ON OUR BENCHMARK DATA

Table 4, Figure 5, and Appendix Table 9 show findings from our evaluation of uncertainty estimation methods on the crafted dataset. By analyzing performance across different uncertainty methods, models, and instruction types, we gain insights into the strengths and limitations of various approaches under both controlled and realistic conditions.

### 3.2.1 COMPARISON OF UNCERTAINTY METHODS: CONTROLLED-EASY AND REALISTIC

**Probe generally outperformed baseline methods in both Controlled-Easy and Realistic version**, as shown in Table 4. Probe consistently outperformed even self-evaluation methods such as berbalized confidence and normalized-p(true). This gap between what the internal states of the model "know" and what they are able to express suggests that their internal layers hold richer, more reliable indicators of uncertainty, which are not fully captured in the model's explicit responses. This points to promising directions for future work, specifically in improving self-evaluation methods by leveraging the rich information within a model's internal representations.

**In Probe, middle layers consistently offer the most informative representations for uncertainty estimation**, particularly in simpler tasks (Controlled-Easy). Appendix Table 10 shows how Probe performance varies across instruction types and layers within the LLMs. The middle layers consistently provide the best signals for uncertainty, indicating that future work could focus on extracting more refined signals from these middle layers to enhance uncertainty estimation methods.

**Self-evaluation methods outperform logit-based ones in Controlled-Easy** In simpler tasks, self-evaluation methods such as verbalized confidence (short answer) and normalized-p(true) (binary choice) consistently outperformed logit-based approaches like perplexity, sequence probability, and entropy. Furthermore, for models like LLaMA2-chat-7B, LLaMA2-chat-13B, and Mistral-7B-Instruct, verbalized confidence performed better than normalized p(true), meaning short-form answer verbalized scores were more calibrated than binary choices in these cases. This overall trend indicates that for Controlled-Easy tasks, models are better at assessing their own correctness using self-evaluation methods compared to logit-based uncertainty estimation.

**Sequence probability excels, but normalized p(true) remains a strong contender in Realistic** In the Realistic evaluation, where models were evaluated on responses with realistic patterns (including length effect), sequence probability performed best across all models, likely due to its ability to exploit the length effect in incorrect responses. However, aside from sequence probability, normalized p(true) consistently ranked as the second-best method across all models, outperforming other logit-based methods like perplexity and mean token entropy. This demonstrates that while sequence probability can take advantage of length bias, normalized p(true) offers a more balanced and reliable uncertainty estimation when length effect is less prominent.

### 3.2.2 COMPARISON OF UNCERTAINTY METHODS: CONTROLLED-HARD

**Probe still struggles with uncertainty estimation in more challenging scenarios.** In Controlled-Hard, Probe AUROC scores drop below 0.60 across all models, revealing that even the internal representations struggle to estimate uncertainty accurately when the task complexity increases. This suggests a limitation in LLMs' ability to handle nuanced or complex uncertainty, highlighting the need for further model fine-tuning or the development of more sophisticated uncertainty estimation methods to improve uncertainty estimation in these difficult tasks.

**Mixed performance between logit-based methods and self-evaluation methods in Controlled-Hard** In the instruction-following samples with more nuanced mistakes, there was a mixed performance between logit-based methods and Normalized p(true). For LLaMA2-chat-7B and LLaMA2-chat-13B, logit-based methods like perplexity and entropy outperformed verbalized confidence and normalized p(true), suggesting that as task complexity increased, logit-based methods became more reliable with those models. However, for Mistral-7B-Instruct and Phi-3-mini, normalized p(true) continued to provide better uncertainty estimation than logit-based methods, demonstrating that the best method can vary depending on the model and its underlying architecture in Controlled-Hard tasks. Uncertainty estimation scores in Controlled-Hard were consistently lower than in other settings and no methods had reliably estimated uncertainty in this setting.

### 3.2.3 COMPARISON OF MODELS

**Mistral-7B-Instruct consistently demonstrates the strongest performance in verbalized confidence across all tasks** (Controlled-Easy, Controlled-Hard, and Realistic), outperforming even the larger LLaMA2-13B model, highlighting its effective internal calibration for self-assessment. On the other hand, **Phi-3-mini-128k leads in normalized p(true)**, consistently achieving the best AUROC across both easy and hard tasks in Controlled, as well as in Realistic versions, showcasing its strength in binary choice settings. In contrast, **LLaMA-2-13B excels in Perplexity**, indicating its proficiency in logit-based uncertainty estimation. These findings are particularly interesting because **smaller models, such as Mistral-7B-Instruct and Phi-3-mini-128k outperform the larger LLaMA-2-chat-13B in self-evaluation methods**. This suggests that factors beyond model size, such as tuning or architecture, may contribute to better uncertainty quantification in certain tasks. However, LLaMA2-13B still shines in logit-based approaches like perplexity, indicating that different methods may favor different models.

### 3.2.4 COMPARISON OF INSTRUCTION TYPES

**The relationship between success rate and AUROC varies significantly across instruction types** As shown in Appendix Table 9, models can correctly follow instructions but still struggle to accurately gauge their own uncertainty, as reflected in the original IFEval evaluations. For instruction types like 'detectable-content' and 'keywords,' there is a clear positive correlation between higher success rates and higher AUROC scores for uncertainty estimation, indicating that when models excel at following instructions, they also tend to estimate uncertainty more reliably. However, this correlation is weaker for other instruction types, such as 'language', 'startend', and 'change-case'.

In particular, 'punctuation' presents a unique challenge. In these tasks, models are instructed to not to use any punctuation in responses. Although this instruction type had the lowest success rates across all models—0.24 (7B), 0.14 (13B), 0.17 (Mistral), and 0.10 (Phi), 'punctuation' yields relatively high AUROC scores, especially in Controlled version with verbalized confidence. This discrepancy may arise because models recognize when they fail to follow the instructions but struggle to adhere to them due to strong priors in their training data. Since most training data includes proper punctuation, instructions to avoid it may conflict with learned patterns, making it difficult for models to follow the instructions while still being somewhat aware of their failure. These findings highlight that while success rates provide some insight into model performance, they do not fully capture the ability to estimate uncertainty in instruction-following.

## 4 RELATED WORK

**Uncertainty estimation in LLMs.** Existing uncertainty estimation methods can be broadly categorized into four types based on the source of information: *verbalized*, *logit-based*, *multi-sample*, and *probing-based* methods. Among them, verbalized methods (Lin et al., 2022; Xiong et al., 2023; Tian et al., 2023) rely on model's self-evaluation by prompting LLMs to explicitly express their uncertainty in theiroutput. Logit-based methods, such as perplexity (Jelinek et al., 1977), sequence probability (Fomicheva et al., 2020), and mean token entropy (Fomicheva et al., 2020) mainly utilize information from the next token prediction distribution. Multi-sample methods (Aichberger et al., 2024; Kuhn et al., 2023; Farquhar et al., 2024) generate multiple responses for the same question, estimating uncertainty through the semantic diversity among the responses. However, these multi-sample methods are less applicable to instruction-following tasks, which only focus on

strict adherence to instructions rather than variations in semantic meaning. Lastly, probing-based methods (Liu et al., 2024; Ahdritz et al., 2024) train external supervised model on model representations to infer uncertainty. In addition, it is worth noting that most existing works focus on factuality-related tasks such as question answering (Xiong et al., 2023; Tian et al., 2023) and summarization tasks (Kuhn et al., 2023), with little attention on instruction-tuning tasks. Our work seek to bridge this gap by evaluating how well current uncertainty metrics capture uncertainties specific to instruction-following scenarios.

**Instruction-following in LLMs** Recent studies have introduced benchmark datasets to evaluate the instruction-following capabilities of LLMs, which includes assessments of general instruction-following (Zhou et al., 2023; Zeng et al., 2023; Qin et al., 2024), refutation tasks (Yan et al., 2024), and format adherence (Xia et al., 2024). Among them, we chose the IFEval dataset (Zhou et al., 2023) as the foundation for our work due to its objective evaluation framework, which uses verifiable instructions to minimize ambiguity in assessment criteria. Additionally, several methods have been explored to enhance instruction-following performance (Zhang et al., 2023a; He et al., 2024; Sun et al., 2024). They highlight the growing interest in LLMs' ability in instruction-following.

**Benchmark datasets for evaluating LLM-as-evaluator** Recent benchmarks (Zeng et al., 2023; Zheng et al., 2024; Zhang et al., 2023b; Wang et al., 2023b) focus on evaluating LLMs' ability to act as evaluators, assessing how well they compare responses in instruction-following. However, our work differs by concentrating on estimating *uncertainty* in instruction-following, comparing various baseline uncertainty estimation methods under controlled conditions. While previous research addresses evaluation biases when LLMs are used as evaluators—such as sensitivity to presentation order (Wang et al., 2023b; Pezeshkpour & Hruschka, 2023) —our benchmark uniquely tackles the challenges posed by entangled factors in evaluating uncertainty estimation ability of LLMs, such as length bias, task execution quality, and varying difficulty levels.

## 5 CONCLUSION

In this paper, we conduct the first comprehensive evaluation of uncertainty estimation in LLMs specifically in the context of instruction-following tasks, addressing a gap in existing research that primarily focuses on fact-based tasks. We identify limitations associated with existing benchmark datasets and introduce a new benchmark with two versions—Controlled and Realistic—designed to provide a comprehensive framework for evaluating uncertainty estimation methods and models under both controlled and real-world conditions. Our analysis revealed that verbalized self-evaluation methods outperform logit-based approaches in Controlled-Easy tasks, while internal model states provide more reliable uncertainty signals in both Controlled-Easy and Realistic settings. However, all methods struggle with more complex tasks in Controlled-Hard, highlighting the limitations of LLMs and future direction for uncertainty estimation in instruction-following.

**Limitations and Future Work** One limitation is the narrow scope of instruction types and domains included in the benchmark, which may not fully capture the diversity of real-world tasks. Furthermore, similar to other research evaluating LLMs, there is a potential risk of leakage, where the models may have been exposed to similar tasks during pre-training, potentially affecting the results. In future work, expanding the benchmark to include a broader range of domains and evaluating more LLMs would furthen deepen understanding of uncertainty estimation in instruction-following. Additional analysis could also investigate *why* LLMs tend to fail to provide accurate uncertainty estimates in instruction-following, which could lead to the development of trustworthy AI agents.

### ACKNOWLEDGMENTS

This work was conducted during an internship at Apple AIML. We sincerely thank Sinead Williamson and Udhay Nallasamy for their valuable feedback and insightful suggestions on this work. We are also grateful to Guillermo Sapiro for his unwavering support and guidance throughout the research.

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

## A APPENDIX

### A.1 DETAILED RESULTS ON IFEVAL DATASET

In the main paper, Table 1 presented the average AUROC scores across all instruction types, providing an overview of how well the baseline uncertainty estimation methods perform in instruction-following tasks. This section provides detailed AUROC results for each individual instruction type in the IFEval dataset in Table 5.

| Model | IFEval Instructions | AUC of uncertainty method | | | | | | SR |
|---|---|---|---|---|---|---|---|---|
| | | Verbal | Ppl | Seq | N-p(t) | p(t) | Ent | |
| LLaMA2-chat-13B | startend | 0.58 | 0.33 | 0.66 | 0.47 | 0.50 | 0.36 | 0.58 |
| | detectable_content | 0.51 | 0.37 | 0.60 | 0.39 | 0.56 | 0.37 | 0.89 |
| | detectable_format | 0.51 | 0.46 | 0.62 | 0.46 | 0.51 | 0.50 | 0.68 |
| | language | 0.51 | 0.49 | 0.61 | 0.48 | 0.53 | 0.52 | 0.58 |
| | change_case | 0.55 | 0.48 | 0.60 | 0.50 | 0.50 | 0.52 | 0.52 |
| | keywords | 0.53 | 0.50 | 0.61 | 0.49 | 0.50 | 0.53 | 0.71 |
| | length_constraints | 0.53 | 0.46 | 0.59 | 0.50 | 0.49 | 0.51 | 0.48 |
| | punctuation | 0.53 | 0.45 | 0.58 | 0.50 | 0.48 | 0.51 | 0.14 |
| | **Average** | 0.53 | 0.44 | 0.61 | 0.47 | 0.51 | 0.48 | 0.57 |
| LLaMA2-chat-7B | startend | 0.55 | 0.30 | 0.56 | 0.57 | 0.52 | 0.32 | 0.67 |
| | detectable_content | 0.48 | 0.25 | 0.55 | 0.48 | 0.47 | 0.29 | 0.85 |
| | detectable_format | 0.55 | 0.47 | 0.49 | 0.51 | 0.52 | 0.47 | 0.66 |
| | language | 0.53 | 0.50 | 0.50 | 0.51 | 0.49 | 0.51 | 0.68 |
| | change_case | 0.52 | 0.49 | 0.55 | 0.54 | 0.52 | 0.50 | 0.48 |
| | keywords | 0.53 | 0.48 | 0.57 | 0.53 | 0.52 | 0.49 | 0.68 |
| | length_constraints | 0.54 | 0.47 | 0.58 | 0.52 | 0.50 | 0.48 | 0.46 |
| | punctuation | 0.54 | 0.47 | 0.57 | 0.53 | 0.50 | 0.49 | 0.24 |
| | **Average** | 0.53 | 0.43 | 0.54 | 0.52 | 0.51 | 0.44 | 0.59 |
| Mistral-7B-Instruct-v0.3 | startend | 0.57 | 0.48 | 0.70 | 0.51 | 0.57 | 0.51 | 0.63 |
| | detectable_content | 0.51 | 0.39 | 0.63 | 0.61 | 0.43 | 0.56 | 0.79 |
| | detectable_format | 0.46 | 0.49 | 0.54 | 0.61 | 0.43 | 0.54 | 0.78 |
| | language | 0.46 | 0.50 | 0.50 | 0.57 | 0.47 | 0.49 | 0.87 |
| | change_case | 0.49 | 0.48 | 0.53 | 0.58 | 0.47 | 0.48 | 0.62 |
| | keywords | 0.50 | 0.48 | 0.52 | 0.57 | 0.47 | 0.47 | 0.73 |
| | length_constraints | 0.50 | 0.49 | 0.53 | 0.56 | 0.47 | 0.49 | 0.55 |
| | punctuation | 0.51 | 0.50 | 0.52 | 0.56 | 0.48 | 0.49 | 0.17 |
| | **Average** | 0.50 | 0.48 | 0.56 | 0.57 | 0.47 | 0.50 | 0.64 |
| Phi-3-mini-128k-instruct | startend | 0.62 | 0.47 | 0.60 | 0.53 | 0.49 | 0.38 | 0.22 |
| | detectable_content | 0.53 | 0.35 | 0.41 | 0.56 | 0.46 | 0.46 | 0.89 |
| | detectable_format | 0.53 | 0.39 | 0.45 | 0.59 | 0.42 | 0.42 | 0.67 |
| | language | 0.49 | 0.43 | 0.47 | 0.53 | 0.41 | 0.48 | 0.97 |
| | change_case | 0.50 | 0.45 | 0.48 | 0.56 | 0.43 | 0.46 | 0.29 |
| | keywords | 0.51 | 0.45 | 0.49 | 0.55 | 0.48 | 0.46 | 0.75 |
| | length_constraints | 0.51 | 0.45 | 0.49 | 0.56 | 0.47 | 0.47 | 0.41 |
| | punctuation | 0.51 | 0.45 | 0.49 | 0.56 | 0.48 | 0.48 | 0.11 |
| | **Average** | 0.53 | 0.43 | 0.48 | 0.55 | 0.45 | 0.45 | 0.54 |

Table 5: **Detailed AUROC for baseline uncertainty estimation methods applied to individual instruction types in the IFEval dataset** (Zhou et al., 2023). This table presents AUROC scores for six uncertainty estimation methods (verbalized confidence, perplexity, sequence probability, normalized p(true), p(true), and Entropy) across four LLMs on all instruction types in the IFEval dataset. The AUROC evaluates how well each method's uncertainty estimates align with the ground truth on correct or incorrect instruction-following. Ppl represents perplexity, Seq stands for sequence probability, N-p(t) is normalized p(true), p(t) refers to p(true), and Ent indicates entropy. Success Rate (SR) represents the model's instruction-following accuracy, as calculated using IFEval's evaluation function. These detailed results reveal variations in performance across instruction types, providing deeper insights into the challenges of estimating uncertainty in instruction-following tasks.

### A.2 PROMPT OF UNCERTAINTY ESTIMATION METHODS

To evaluate uncertainty in instruction-following tasks, we employed several baseline methods, including self-evaluation methods (verbalized confidence, normalized p(true), and p(true)) and logits-based methods (perplexity, sequence probability, and mean token entropy). For self-evaluation meth-

ods, we used specific prompts to engage the models in assessing their own uncertainty. Below, we provide the exact prompts used in our experiments for both verbalized confidence and p(true) methods.

**Prompt for Verbalized Confidence** is used to assess the model's self-reported confidence, asking the model to score its perceived likelihood that its response follows the given instructions. The model is instructed to provide a confidence score ranging from 0 (no confidence) to 9 (high confidence).

**Prompt for p(true) and Normalized p(true)** evaluates the model's probability estimate of whether its previous response correctly followed the instructions. The model is prompted to assign probabilities to a binary choice, with p(true) reflecting the likelihood that the response is correct. Normalized p(true) adjusts the raw probabilities of both options to account for biases, normalizing the confidence scores.

---

**Prompt for verbalized confidence**

You are a helpful assistant in evaluating the quality of the outputs for a given instruction. Your goal is to score a given output for the given instruction.

Score the output for the given instruction. The output is generated by an AI chatbot. You should give an overall score (an integer) on a scale of 0 to 9, where a higher score indicates better overall performance.

Do NOT provide any explanation for your evaluation. Your response should be ONLY the score, an integer between 0 and 9.

# Instruction: {input}
# Output: {instruction following response}
# Score of the Output (Your response should be ONLY the score, an integer between 0 and 9):

---

**Prompt for p(true) and Normalized p(true)**

You are a helpful assistant in evaluating the quality of the outputs for a given instruction. Your goal is to score a given output for the given instruction.

Question: {instruction input}
Response: {instruction following response}
Does the response: (A) Follow instructions (B) Not follow instructions.
The answer is: (

---

## A.3 PROMPTS FOR GENERATING OUR CONTROLLED BENCHMARK DATASET

To create the Controlled version of our benchmark dataset, we employed carefully designed prompts that generate responses with controlled token lengths, varying difficulty levels, and high task quality. The aim was to ensure that the evaluation focuses solely on uncertainty from instruction-following, without interference from factors such as task execution quality or response length. Below, we provide the full prompts used to generate both subtle off-target and completely wrong responses, as well as the prompt for scoring task quality or instruction-following.

---

**Prompt for generating subtle off-target responses**

I want to generate synthetic data to test LLM's ability to evaluate instruction following. There are three types of responses in instruction-following; correct, subtly off-target, totally wrong. Your task involves generating subtly off-target responses that are subtly disregard the instruction. Subtly off-target response should be classified as false according to pre-defined evaluation functions based on evaluation focus provided below.

For example, if the instruction is "Make sure to include at least 10 placeholders..." and kwargs is "number-placeholders 10" then subtly off-target response might include only 8 placeholders. Another example, if the instruction is "Make sure to include keywords: love, hate" and kwargs is "keywords:love, hate", then subtly off-target response may include only one of them.

Make sure the length of your generated subtly off-target response matches the length of the correct response provided below, aiming for a similar level of detail and complexity.

\# Instruction: {Instruction-and-task input}
\# Evaluation focus: {Type}
\# Details of evaluation criterion: {Kwargs}
\# Correctly followed response: {Correct Response}
\# Subtly off-target response:

---

**Prompt for generating totally wrong responses**

I want to generate synthetic data to test LLM's ability to evaluate instruction following. There are three types of responses in instruction-following; correct, confusingly wrong, totally wrong. Your task involves generating totally wrong responses that are clearly disregards the instruction. Your totally wrong response should be classified as false according to pre-defined evaluation functions based on evaluation focus provided below.

For example, if the instruction is "Make sure to include at least 10 placeholders..." and kwargs is "number-placeholders:10" then wrong response might include only 2 placeholders. Another example, if the instruction is "Make sure to include keywords: love, hate" and kwargs is "keywords:love, hate", then wrong response may include none of them.

Make sure the length of your generated confusing response matches the length of the correct response provided below, aiming for a similar level of detail and complexity.

\# Instruction: {Instruction-and-task input}
\# Evaluation focus: {Type}
\# Details of evaluation criterion: {Kwargs}
\# Correctly followed response: {Correct Response}
\# Confusingly wrong response:

---

---

**Prompt for generating subtle off-target responses**

{Instruction-and-task input}

---

**Prompt for scoring task quality**

You are a helpful assistant in evaluating the quality of the outputs for a given instruction. Your goal is to score a given output for the given instruction. You should give an overall score (an integer) on a scale of 0 to 9, where a higher score indicates better overall performance. Do NOT provide any explanation for your evaluation.

# Instruction: {Task-only-input}
# Output:{Response}
# Score of the Output (Your response should be ONLY the score, an integer between 0 and 9):

---

**Prompt for scoring instruction following**

You are a helpful assistant in evaluating how well a given response follows a given instruction. Your goal is to score a given output for the given instruction. You should give an overall score (an integer) on a scale of 0 to 9, where a higher score indicates better overall performance. Do NOT provide any explanation for your evaluation.

# Instruction: {Instruction-only-input}
# Output:{Response}
# Score of the Output (Your response should be ONLY the score, an integer between 0 and 9):

---

## A.4 WHY DO WE CHOOSE IFEVAL DATASET?

Here, we would like to emphasize why we choose IFEval as our primary dataset instead of using real-world dataset with different contexts and domains.

First, we select IFEval to focus on our scope which is 'single, simple, and non-ambiguous instructions'. Real-world datasets often involve complex, ambiguous, or multi-instruction prompts, which can conflate multiple factors affecting uncertainty. For this first systematic evaluation, we chose to focus on single, simple, and verifiable instructions to ensure clarity and isolate the uncertainty estimation process. The IFEval dataset is well-suited for this purpose, as it provides 25 distinct types of simple and clear instructions that align with our goal of establishing a robust baseline.

Second, we want to avoid evaluator-induced uncertainties. Most real-world tasks and benchmark datasets rely on LLM-based evaluators to determine whether a response follows an instruction. However, LLM-based evaluators may introduce their own uncertainties or make errors in assessing success or failure, which could obscure the true uncertainty estimation capabilities of the tested models. The IFEval dataset avoids this issue by including instructions with deterministic evaluation programs that objectively verify compliance. For instance, an instruction like "please do not include keywords: ..." can be automatically validated using a simple program to check for the presence of those keywords. This feature eliminates ambiguity in evaluation and allows us to directly focus on LLMs' ability to estimate uncertainty.

Our main contribution is the careful design of benchmark data specifically tailored to 'uncertainty estimation' in instruction-following contexts. We believe that those underlying considerations (e.g., length bias, task quality entanglement, and varying levels of model difficulty) presented in our study can extend to future instruction-following datasets. While IFEval serves as an ideal starting point for this research, we hope our framework inspires future efforts to tackle uncertainty estimation in more complex, real-world tasks.

## A.5 IFEval DATA EXAMPLES

Table 6 presents examples from the IFEval dataset, such as tasks like writing a resume or creating a joke about programmers. The instructions assigned to each task vary, requiring the model to follow specific guidelines such as including or excluding certain keywords, ensuring word usage meets a specific frequency, and adhering to formatting rules. The keywords that must be included or excluded differ based on the task. For instance, in the resume task, keywords might include "resume", "software", or "engineer", whereas in the joke task, the focus may shift to terms like "syntax" or "code". These varied instructions introduce diverse challenges for the model in instruction-following.

| Type | Example |
|---|---|
| **Task** | *Write a resume for a software engineer with 5+ years of experience in the Bay Area, CA.* |
| **Instruction** | **keywords:existence**
Make sure to include the keywords: "skills", "technology", "career".

**keywords:forbidden**
Do not include the following keywords: resume, software, engineer, experience.

**keywords:frequency**
Make sure to use the word "qualifications" at least 2 times.

**startend:end checker**
Your resume must end with the exact phrase "Looking forward to contributing to innovative projects."

**detectable content:number placeholders**
Make sure to include at least 5 placeholders represented by square brackets, such as [name]. |
| **Task** | *Write a joke about programmers.* |
| **Instruction** | **keywords:existence**
Make sure to include the keywords: "humor", "code", "life".

**keywords:forbidden**
Do not include the following keywords: joke, programmers.

**keywords:frequency**
Make sure to use the word "syntax" at least 3 times.

**startend:end checker**
Your programmer joke must end with the exact phrase "And that's the real bug in the code of life."

**detectable content:number placeholders**
Make sure to include at least 3 placeholders represented by square brackets, such as [name]. |

Table 6: **Examples from the IFEval dataset.** This table shows two tasks: writing a resume and crafting a joke about programmers. Each task is paired with multiple instruction types, such as including/excluding keywords, ensuring word frequency, and adhering to specific content formatting rules.

## A.6 STATS OF OUR BENCHMARK DATA

This section provides comprehensive statistics for our benchmark datasets, detailing the number of data points and token length distributions. These statistics illustrate the construction and characteristics of both the Controlled and Realistic versions of our benchmark dataset.

Figures 3a and Figures 3b show the token length distributions for the Controlled and Realistic datasets, respectively, highlighting the absence of length bias in the Controlled version and the presence of natural length variation in the Realistic version. Figures 4a and Figures 4b display the model contribution to total correct and incorrect responses in the Realistic version, ensuring diverse data sources. Table 7 presents the number of correct and incorrect responses in both the Controlled and Realistic versions. Table 8 provides the mean token lengths across different instruction types, underscoring the careful balancing of token lengths in the Controlled version.

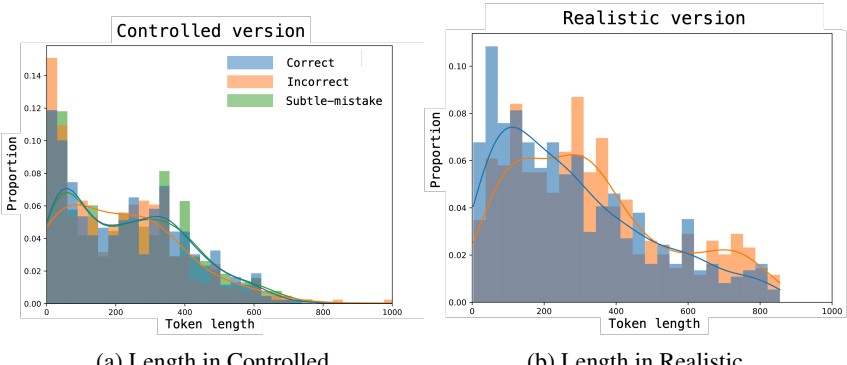

(a) Length in Controlled

(b) Length in Realistic

Figure 3: **Token length distributions for the Controlled and Realistic versions of our benchmark dataset**. The distributions are normalized by the total number of responses in each class. (a) Token length distribution in the Controlled version, where token lengths are carefully balanced between correct and incorrect responses. (b) Token length distribution in the Realistic version, where token length reflects the natural variability of model-generated responses.

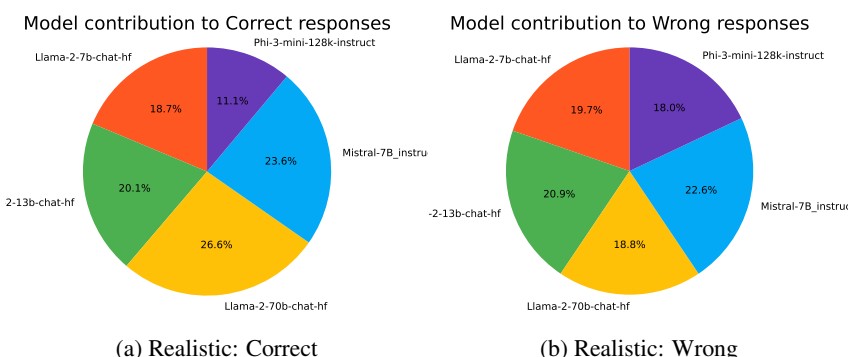

(a) Realistic: Correct

(b) Realistic: Wrong

Figure 4: **Model contributions to the Realistic version of the benchmark dataset**. (a) Pie chart representing the model contribution to correct responses. (b) Pie chart representing the model contribution to incorrect responses. These contributions ensure that the Realistic dataset captures diverse responses from various LLMs.

| | Controlled | | | Realistic | |
|---|---|---|---|---|---|
| | correct | wrong | subtle off-target | wrong | correct |
| startend | 58 | 49 | 58 | 49 | 60 |
| detectable_content | 43 | 46 | 45 | 21 | 15 |
| detectable_format | 127 | 111 | 107 | 123 | 130 |
| language | 27 | 28 | 14 | 5 | 7 |
| change_case | 61 | 78 | 74 | 60 | 61 |
| keywords | 127 | 134 | 109 | 102 | 108 |
| length_constraints | 95 | 107 | 108 | 94 | 92 |
| punctuation | 40 | 56 | 52 | 20 | 12 |
| Sum | 578 | 609 | 567 | 474 | 485 |
| Total | 429 | 411 | 381 | 345 | 369 |

Table 7: **Summary of the number of data points for correct and incorrect cases in both the Controlled and Realistic versions**. One example may contain multiple instructions, so the total number of examples is less than the sum of data points across all instruction types.

| | Controlled | | | Realistic | |
|---|---|---|---|---|---|
| | correct | wrong | confusing | wrong | correct |
| startend | 242.81 | 216.29 | 267.47 | 411.08 | 243.03 |
| detectable_content | 282.72 | 270.15 | 272.27 | 366.86 | 296.20 |
| detectable_format | 244.69 | 244.16 | 233.14 | 349.14 | 320.44 |
| language | 91.37 | 115.75 | 144.57 | 238.00 | 140.29 |
| change_case | 229.97 | 291.04 | 247.73 | 273.82 | 288.25 |
| keywords | 232.47 | 247.34 | 247.34 | 315.70 | 254.56 |
| length_constraints | 260.89 | 317.97 | 286.83 | 336.82 | 285.45 |
| punctuation | 142.50 | 182.54 | 177.81 | 173.70 | 50.75 |
| Average | 215.93 | 235.65 | 234.64 | 308.14 | 234.87 |

Table 8: **Mean token length for each instruction type in both the Controlled and Realistic versions of the dataset**. The Controlled version neutralizes the impact of token length, while the Realistic version reflects natural length variation.

A.7 MORE RESULTS ON OUR BENCHMARK DATA

In this section, we provide additional results on uncertainty estimation performance using our crafted benchmark datasets. These results offer deeper insights into how different LLMs and uncertainty estimation methods perform across various evaluation scenarios, both controlled and realistic. The radar charts and tables present a detailed comparison of models' performance, using AUROC averaged across instruction types for different uncertainty estimation methods. Figure 5 and Figure 6 shows the radar charts on IFEval data and our benchmark data. Table 9 provides detailed AUROC scores for each instruction type.

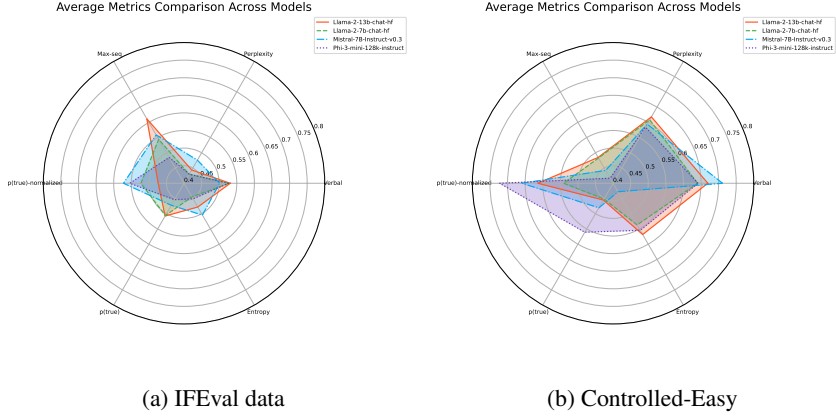

(a) IFEval data          (b) Controlled-Easy

Figure 5: **Model comparison** of uncertainty estimation across different evaluation scenarios. Radar charts illustrate the performance of four LLMs on six uncertainty metrics, with AUROC averaged across instruction types. (a) Results based on IFEval data. (b) Results on Controlled-Easy version of our crafted data (distinguishing correct and incorrect responses).

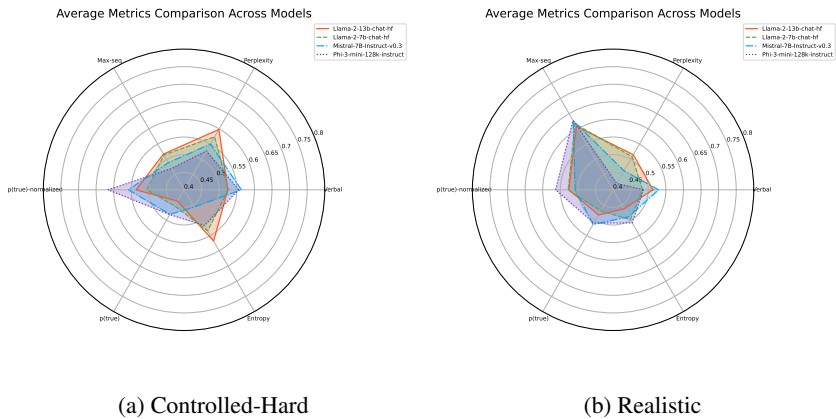

(a) Controlled-Hard          (b) Realistic

Figure 6: **Model comparison** of uncertainty estimation across different evaluation scenarios. Radar charts illustrate the performance of four LLMs on six uncertainty metrics, with AUROC averaged across instruction types. (a) AUROC on the Controlled-Hard (distinguishing correct and subtle off-target responses) and (b) AUROC on the Realistic version of our crafted data.

| Model | Instructions | Controlled-Easy | | | | | | Controlled-Hard | | | | | | Realistic | | | | | |
|---|---|---|---|---|---|---|---|---|---|---|---|---|---|---|---|---|---|---|---|
| | | Verbal | Ppl | Seq | N-p(t) | p(t) | Ent | Verbal | Ppl | Seq | N-p(t) | p(t) | Ent | Verbal | Ppl | Seq | N-p(t) | p(t) | Ent |
| LLaMA2-chat-13B | startend | 0.65 | 0.58 | 0.50 | 0.67 | 0.44 | 0.58 | 0.52 | 0.58 | 0.54 | 0.56 | 0.43 | 0.59 | 0.52 | 0.49 | 0.67 | 0.53 | 0.54 | 0.40 |
| | detectable_content | 0.67 | 0.63 | 0.49 | 0.64 | 0.48 | 0.59 | 0.50 | 0.59 | 0.51 | 0.54 | 0.44 | 0.56 | 0.52 | 0.53 | 0.67 | 0.54 | 0.52 | 0.44 |
| | detectable_format | 0.66 | 0.62 | 0.48 | 0.59 | 0.45 | 0.59 | 0.51 | 0.61 | 0.52 | 0.53 | 0.43 | 0.58 | 0.51 | 0.53 | 0.62 | 0.54 | 0.46 | 0.48 |
| | language | 0.67 | 0.64 | 0.46 | 0.59 | 0.44 | 0.60 | 0.53 | 0.62 | 0.51 | 0.53 | 0.42 | 0.57 | 0.50 | 0.52 | 0.62 | 0.53 | 0.46 | 0.47 |
| | change_case | 0.66 | 0.62 | 0.47 | 0.59 | 0.46 | 0.56 | 0.51 | 0.60 | 0.50 | 0.51 | 0.43 | 0.57 | 0.51 | 0.51 | 0.58 | 0.51 | 0.46 | 0.47 |
| | keywords | 0.68 | 0.62 | 0.49 | 0.61 | 0.46 | 0.56 | 0.53 | 0.60 | 0.51 | 0.53 | 0.45 | 0.56 | 0.52 | 0.51 | 0.57 | 0.51 | 0.47 | 0.48 |
| | length_constraints | 0.68 | 0.61 | 0.50 | 0.62 | 0.46 | 0.55 | 0.55 | 0.59 | 0.52 | 0.55 | 0.45 | 0.56 | 0.51 | 0.51 | 0.57 | 0.52 | 0.48 | 0.48 |
| | punctuation | 0.68 | 0.61 | 0.50 | 0.61 | 0.46 | 0.56 | 0.54 | 0.59 | 0.52 | 0.54 | 0.45 | 0.56 | 0.51 | 0.52 | 0.57 | 0.52 | 0.48 | 0.48 |
| | **Average** | 0.67 | 0.62 | 0.48 | 0.61 | 0.46 | 0.57 | 0.52 | 0.60 | 0.52 | 0.54 | 0.44 | 0.57 | 0.51 | 0.52 | 0.61 | 0.53 | 0.48 | 0.46 |
| LLaMA2-chat-7B | startend | 0.64 | 0.59 | 0.50 | 0.55 | 0.43 | 0.55 | 0.52 | 0.57 | 0.54 | 0.49 | 0.43 | 0.55 | 0.41 | 0.52 | 0.74 | 0.54 | 0.44 | 0.46 |
| | detectable_content | 0.65 | 0.63 | 0.49 | 0.53 | 0.41 | 0.56 | 0.52 | 0.58 | 0.52 | 0.49 | 0.43 | 0.55 | 0.43 | 0.55 | 0.72 | 0.52 | 0.42 | 0.50 |
| | detectable_format | 0.61 | 0.62 | 0.48 | 0.53 | 0.44 | 0.57 | 0.51 | 0.58 | 0.52 | 0.51 | 0.45 | 0.54 | 0.50 | 0.50 | 0.60 | 0.54 | 0.48 | 0.51 |
| | language | 0.63 | 0.62 | 0.46 | 0.52 | 0.43 | 0.54 | 0.51 | 0.59 | 0.51 | 0.50 | 0.45 | 0.52 | 0.49 | 0.50 | 0.61 | 0.53 | 0.48 | 0.50 |
| | change_case | 0.63 | 0.60 | 0.47 | 0.54 | 0.46 | 0.52 | 0.52 | 0.57 | 0.50 | 0.51 | 0.45 | 0.53 | 0.50 | 0.49 | 0.57 | 0.53 | 0.48 | 0.50 |
| | keywords | 0.65 | 0.61 | 0.49 | 0.55 | 0.48 | 0.52 | 0.54 | 0.57 | 0.51 | 0.52 | 0.47 | 0.52 | 0.51 | 0.50 | 0.57 | 0.53 | 0.49 | 0.50 |
| | length_constraints | 0.66 | 0.60 | 0.49 | 0.56 | 0.48 | 0.51 | 0.55 | 0.57 | 0.51 | 0.52 | 0.47 | 0.53 | 0.51 | 0.50 | 0.56 | 0.51 | 0.48 | 0.50 |
| | punctuation | 0.65 | 0.60 | 0.50 | 0.56 | 0.49 | 0.52 | 0.55 | 0.57 | 0.51 | 0.51 | 0.47 | 0.53 | 0.51 | 0.50 | 0.57 | 0.51 | 0.48 | 0.51 |
| | **Average** | 0.64 | 0.61 | 0.48 | 0.54 | 0.45 | 0.54 | 0.53 | 0.57 | 0.51 | 0.51 | 0.45 | 0.53 | 0.48 | 0.51 | 0.62 | 0.53 | 0.47 | 0.50 |
| Mistral-7B-Instruct-v0.3 | startend | 0.69 | 0.59 | 0.46 | 0.64 | 0.46 | 0.47 | 0.54 | 0.54 | 0.51 | 0.56 | 0.43 | 0.46 | 0.59 | 0.40 | 0.76 | 0.52 | 0.53 | 0.51 |
| | detectable_content | 0.70 | 0.60 | 0.43 | 0.65 | 0.46 | 0.46 | 0.53 | 0.56 | 0.49 | 0.54 | 0.45 | 0.48 | 0.53 | 0.43 | 0.72 | 0.49 | 0.53 | 0.53 |
| | detectable_format | 0.71 | 0.63 | 0.44 | 0.66 | 0.48 | 0.45 | 0.56 | 0.57 | 0.49 | 0.57 | 0.48 | 0.48 | 0.53 | 0.48 | 0.62 | 0.52 | 0.51 | 0.49 |
| | language | 0.71 | 0.61 | 0.41 | 0.67 | 0.49 | 0.41 | 0.56 | 0.55 | 0.48 | 0.56 | 0.49 | 0.44 | 0.53 | 0.47 | 0.62 | 0.51 | 0.51 | 0.48 |
| | change_case | 0.70 | 0.57 | 0.43 | 0.65 | 0.49 | 0.42 | 0.56 | 0.54 | 0.48 | 0.54 | 0.50 | 0.45 | 0.50 | 0.46 | 0.58 | 0.51 | 0.51 | 0.46 |
| | keywords | 0.72 | 0.58 | 0.45 | 0.66 | 0.49 | 0.40 | 0.58 | 0.54 | 0.49 | 0.55 | 0.50 | 0.45 | 0.52 | 0.48 | 0.58 | 0.51 | 0.51 | 0.47 |
| | length_constraints | 0.73 | 0.58 | 0.45 | 0.68 | 0.48 | 0.41 | 0.59 | 0.55 | 0.49 | 0.57 | 0.50 | 0.45 | 0.52 | 0.49 | 0.57 | 0.50 | 0.51 | 0.48 |
| | punctuation | 0.73 | 0.58 | 0.46 | 0.67 | 0.49 | 0.41 | 0.58 | 0.55 | 0.49 | 0.57 | 0.49 | 0.45 | 0.52 | 0.49 | 0.57 | 0.51 | 0.50 | 0.48 |
| | **Average** | 0.71 | 0.59 | 0.44 | 0.66 | 0.48 | 0.43 | 0.56 | 0.55 | 0.49 | 0.56 | 0.48 | 0.46 | 0.53 | 0.46 | 0.63 | 0.51 | 0.51 | 0.49 |
| Phi-3-mini-128k-instruct | startend | 0.62 | 0.59 | 0.47 | 0.74 | 0.53 | 0.56 | 0.56 | 0.55 | 0.46 | 0.64 | 0.40 | 0.50 | 0.47 | 0.39 | 0.70 | 0.65 | 0.48 | 0.48 |
| | detectable_content | 0.66 | 0.64 | 0.40 | 0.71 | 0.55 | 0.58 | 0.53 | 0.55 | 0.44 | 0.59 | 0.43 | 0.49 | 0.47 | 0.40 | 0.65 | 0.61 | 0.48 | 0.50 |
| | detectable_format | 0.64 | 0.57 | 0.41 | 0.73 | 0.53 | 0.57 | 0.55 | 0.52 | 0.48 | 0.62 | 0.47 | 0.53 | 0.48 | 0.43 | 0.64 | 0.55 | 0.52 | 0.52 |
| | language | 0.63 | 0.58 | 0.41 | 0.72 | 0.55 | 0.57 | 0.55 | 0.52 | 0.47 | 0.61 | 0.49 | 0.52 | 0.49 | 0.42 | 0.65 | 0.56 | 0.51 | 0.52 |
| | change_case | 0.62 | 0.56 | 0.41 | 0.70 | 0.57 | 0.55 | 0.55 | 0.50 | 0.47 | 0.59 | 0.51 | 0.53 | 0.48 | 0.44 | 0.63 | 0.54 | 0.53 | 0.52 |
| | keywords | 0.65 | 0.59 | 0.39 | 0.72 | 0.59 | 0.54 | 0.56 | 0.52 | 0.48 | 0.61 | 0.52 | 0.52 | 0.50 | 0.42 | 0.60 | 0.53 | 0.53 | 0.50 |
| | length_constraints | 0.65 | 0.58 | 0.41 | 0.73 | 0.58 | 0.53 | 0.57 | 0.53 | 0.48 | 0.61 | 0.52 | 0.52 | 0.50 | 0.44 | 0.58 | 0.53 | 0.52 | 0.51 |
| | punctuation | 0.65 | 0.57 | 0.43 | 0.73 | 0.58 | 0.54 | 0.57 | 0.53 | 0.48 | 0.63 | 0.52 | 0.00 | 0.50 | 0.43 | 0.58 | 0.54 | 0.52 | 0.51 |
| | **Average** | 0.64 | 0.58 | 0.42 | 0.72 | 0.56 | 0.55 | 0.55 | 0.53 | 0.47 | 0.62 | 0.48 | 0.52 | 0.49 | 0.42 | 0.63 | 0.56 | 0.51 | 0.51 |

Table 9: **AUC for each instruction type** for different LLMs and uncertainty estimation methods across three settings. The uncertainty estimation methods include Verbalized confidence (Verb), Perplexity (Ppl), Sequence probability (Seq), Normalized p(true) (N-p(t)), p(true), Entropy (Ent), and linear probing on internal states (Probe). Bold values indicate the best-performing method for each model and condition, while underlined values denote the second-best performing method.

## A.8 More results on internal states of LLMs

This section presents additional results on the effectiveness of linear probing (**Probe**) on the internal states of different LLMs across various instruction types. As outlined in the main paper, we applied linear models to early, middle, and late layers of each model to determine how well internal states capture uncertainty in instruction-following tasks.

Table 10 provides the AUROC performance of probes across different instruction types and layers for several models. Results are shown for both linear and projection-based methods. Figure 7, Figure 8, and Figure 9 visualizes these results through heatmaps, showing performance across early, middle, and late layers for each instruction type.

| Model | Instructions | Controlled-Easy | | | Controlled-Hard | | | Realistic | | |
|---|---|---|---|---|---|---|---|---|---|---|
| | | Early | Middle | Last | Early | Middle | Last | Early | Middle | Last |
| LLaMA-2-chat-13B | startend | 0.74 | 0.88 | 0.81 | 0.53 | 0.48 | 0.39 | 0.74 | 0.83 | 0.80 |
| | detectable_content | 0.83 | 0.86 | 0.81 | 0.74 | 0.57 | 0.52 | 0.63 | 0.44 | 0.56 |
| | detectable_format | 0.87 | 0.84 | 0.84 | 0.57 | 0.63 | 0.51 | 0.54 | 0.58 | 0.51 |
| | language | 0.98 | 0.84 | 0.97 | 0.97 | 0.89 | 0.81 | 1.00 | 1.00 | 0.50 |
| | change_case | 0.77 | 0.69 | 0.63 | 0.46 | 0.50 | 0.45 | 0.55 | 0.39 | 0.51 |
| | keywords | 0.78 | 0.80 | 0.82 | 0.53 | 0.64 | 0.59 | 0.54 | 0.55 | 0.46 |
| | length_constraints | 0.83 | 0.80 | 0.71 | 0.62 | 0.57 | 0.54 | 0.63 | 0.59 | 0.56 |
| | punctuation | 0.58 | 0.62 | 0.54 | 0.51 | 0.34 | 0.36 | 0.78 | 0.75 | 1.00 |
| | **Average** | 0.80 | 0.79 | 0.76 | 0.62 | 0.58 | 0.52 | 0.68 | 0.64 | 0.61 |
| LLaMA-2-chat-7B | startend | 0.86 | 0.73 | 0.56 | 0.50 | 0.56 | 0.44 | 0.84 | 0.85 | 0.70 |
| | detectable_content | 0.88 | 0.92 | 0.88 | 0.47 | 0.45 | 0.45 | 0.93 | 0.73 | 0.44 |
| | detectable_format | 0.74 | 0.73 | 0.71 | 0.53 | 0.50 | 0.48 | 0.57 | 0.58 | 0.59 |
| | language | 0.76 | 0.75 | 0.71 | 0.86 | 0.72 | 0.58 | 0.33 | 0.67 | 0.50 |
| | change_case | 0.63 | 0.51 | 0.54 | 0.41 | 0.49 | 0.56 | 0.60 | 0.60 | 0.58 |
| | keywords | 0.58 | 0.67 | 0.70 | 0.50 | 0.44 | 0.48 | 0.56 | 0.61 | 0.46 |
| | length_constraints | 0.77 | 0.75 | 0.75 | 0.54 | 0.51 | 0.55 | 0.53 | 0.42 | 0.51 |
| | punctuation | 0.51 | 0.67 | 0.55 | 0.39 | 0.44 | 0.49 | 0.44 | 0.44 | 1.00 |
| | **Average** | 0.72 | 0.72 | 0.67 | 0.52 | 0.51 | 0.50 | 0.60 | 0.61 | 0.60 |
| Mistral-7B-Instruct-v0.3 | startend | 0.80 | 0.79 | 0.76 | 0.33 | 0.43 | 0.54 | 0.54 | 0.53 | 0.76 |
| | detectable_content | 0.67 | 0.80 | 0.88 | 0.47 | 0.21 | 0.38 | 0.81 | 0.63 | 0.60 |
| | detectable_format | 0.75 | 0.75 | 0.79 | 0.58 | 0.68 | 0.51 | 0.52 | 0.59 | 0.59 |
| | language | 0.98 | 0.83 | 0.88 | 0.94 | 0.94 | 0.81 | 1.00 | 1.00 | 0.59 |
| | keywords | 0.57 | 0.59 | 0.66 | 0.61 | 0.50 | 0.58 | 0.44 | 0.48 | 0.43 |
| | change_case | 0.73 | 0.87 | 0.85 | 0.49 | 0.58 | 0.58 | 0.59 | 0.66 | 0.58 |
| | length_constraints | 0.82 | 0.86 | 0.84 | 0.61 | 0.73 | 0.65 | 0.49 | 0.60 | 0.53 |
| | punctuation | 0.47 | 0.48 | 0.49 | 0.43 | 0.41 | 0.56 | 0.78 | 0.78 | 0.88 |
| | **Average** | 0.73 | 0.75 | 0.77 | 0.56 | 0.56 | 0.58 | 0.65 | 0.66 | 0.62 |
| Phi-3-mini-128k-instruct | startend | 0.68 | 0.82 | 0.78 | 0.48 | 0.47 | 0.45 | 0.80 | 0.89 | 0.93 |
| | detectable_content | 0.78 | 0.84 | 0.95 | 0.47 | 0.45 | 0.45 | 0.93 | 0.81 | 0.92 |
| | detectable_format | 0.78 | 0.82 | 0.83 | 0.55 | 0.61 | 0.68 | 0.50 | 0.52 | 0.49 |
| | language | 0.72 | 0.97 | 0.90 | 0.54 | 0.56 | 0.86 | 1.00 | 1.00 | 0.67 |
| | change_case | 0.55 | 0.60 | 0.61 | 0.54 | 0.50 | 0.63 | 0.46 | 0.51 | 0.46 |
| | keywords | 0.61 | 0.88 | 0.85 | 0.43 | 0.46 | 0.65 | 0.72 | 0.49 | 0.43 |
| | length_constraints | 0.58 | 0.84 | 0.85 | 0.60 | 0.60 | 0.58 | 0.60 | 0.62 | 0.73 |
| | punctuation | 0.65 | 0.57 | 0.47 | 0.49 | 0.64 | 0.47 | 0.44 | 0.89 | 0.67 |
| | **Average** | 0.67 | 0.79 | 0.78 | 0.51 | 0.53 | 0.60 | 0.68 | 0.72 | 0.66 |

Table 10: **AUROC performance of linear probing (Probe) applied to internal states** of early, middle, and late layers for various models and instruction types. Results are reported for both linear and projection methods, showing that middle layers generally offer more informative representations for uncertainty estimation.

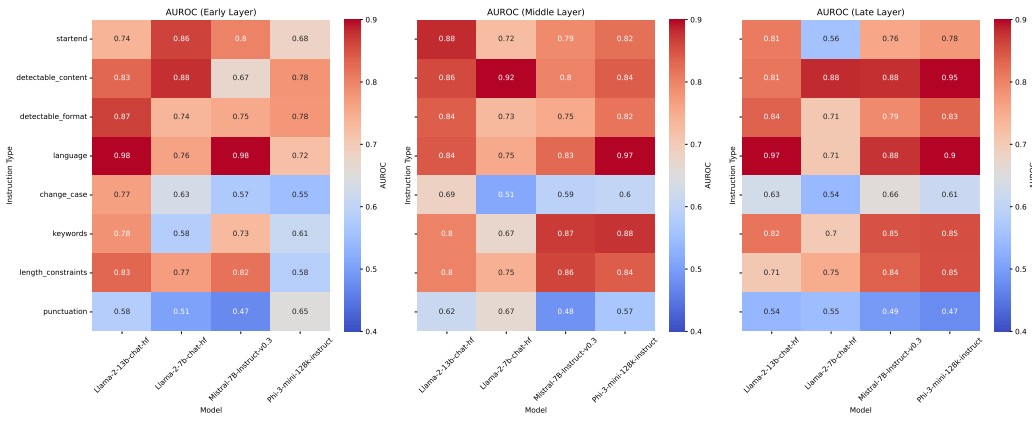

Figure 7: Controlled-Easy: Heatmaps showing the AUROC performance of probes on early, middle, and late layers for different instruction types across various LLMs.

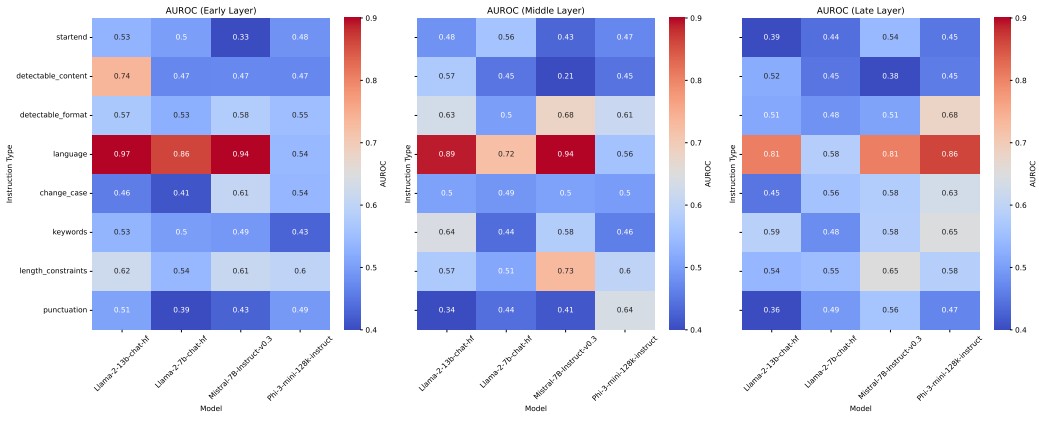

Figure 8: Controlled-Hard: Heatmaps showing the AUROC performance of probes on early, middle, and late layers for different instruction types across various LLMs.

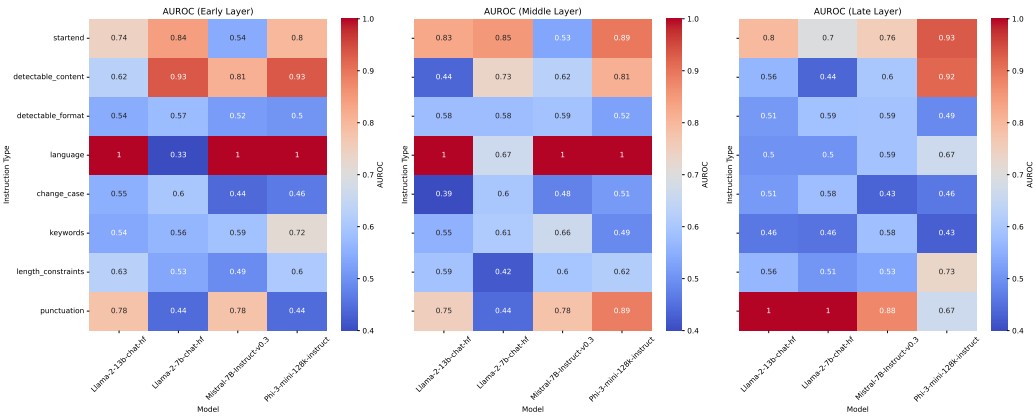

Figure 9: Realistic version: Heatmaps showing the AUROC performance of probes on early, middle, and late layers for different instruction types across various LLMs.

## A.9 LENGTH EFFECT MODELS

This section analyzes the impact of token length on uncertainty estimation across different instruction types and models. We examine the relationship between token length and correctness in both correct and incorrect responses, revealing a prevalent length signal in existing instruction-following datasets like IFEval. This effect can introduce biases in uncertainty estimation methods that rely on token length, complicating accurate assessments of model performance.

Table 11 presents the mean and standard deviation of token lengths for both correct and incorrect responses across four LLMs. The results show that, on average, incorrect responses are consistently longer than correct ones, further underscoring the influence of token length on uncertainty estimation. Figure 10 provides a visual representation of the token length distributions for three models: LLaMA-2-chat-13B, Mistral-7B-Inst-v0.3, and Phi-3-128k-inst. The figure illustrates how incorrect responses tend to be longer than correct responses, reinforcing the presence of a length signal in naturally generated datasets like IFEval(Zhou et al., 2023).

| Length of responses (token) | Llama-2-7b-chat-hf | | Llama-2-13b-chat-hf | | Mistral-7B-Inst-v0.3 | | Phi-3-mini-128k-instruct | |
| | correct | wrong | correct | wrong | correct | wrong | correct | wrong |
|---|---|---|---|---|---|---|---|---|
| startend | $210.03 \pm 146.93$ | $314.59 \pm 185.24$ | $221.00 \pm 154.75$ | $338.29 \pm 198.16$ | $211.67 \pm 151.23$ | $284.67 \pm 187.17$ | $614.00 \pm 206.95$ | $634.28 \pm 226.97$ |
| detectable_content | $269.54 \pm 129.69$ | $400.63 \pm 176.84$ | $276.65 \pm 156.09$ | $376.05 \pm 124.84$ | $282.20 \pm 147.75$ | $309.71 \pm 151.69$ | $593.28 \pm 188.23$ | $582.24 \pm 196.44$ |
| detectable_format | $279.28 \pm 201.31$ | $305.40 \pm 180.85$ | $316.56 \pm 203.45$ | $309.54 \pm 173.21$ | $263.00 \pm 191.05$ | $246.26 \pm 176.43$ | $619.03 \pm 213.78$ | $610.06 \pm 220.98$ |
| language | $159.86 \pm 118.60$ | $192.29 \pm 100.09$ | $137.14 \pm 81.71$ | $204.71 \pm 103.24$ | $155.25 \pm 81.34$ | $119.50 \pm 74.39$ | $249.20 \pm 152.07$ | $240.09 \pm 176.64$ |
| change_case | $210.77 \pm 138.66$ | $338.36 \pm 179.42$ | $265.11 \pm 174.03$ | $270.75 \pm 122.33$ | $202.49 \pm 150.06$ | $296.98 \pm 179.04$ | $569.87 \pm 152.97$ | $601.41 \pm 206.22$ |
| keywords | $272.45 \pm 179.84$ | $296.84 \pm 164.83$ | $282.40 \pm 188.00$ | $280.42 \pm 172.82$ | $280.89 \pm 200.39$ | $249.15 \pm 164.24$ | $600.66 \pm 238.87$ | $590.06 \pm 235.95$ |
| length_constraints | $262.62 \pm 205.11$ | $334.77 \pm 174.38$ | $294.72 \pm 233.91$ | $334.29 \pm 174.02$ | $296.21 \pm 236.77$ | $333.06 \pm 185.42$ | $546.00 \pm 258.54$ | $639.02 \pm 208.89$ |
| punctuation | $49.20 \pm 42.39$ | $245.63 \pm 175.68$ | $46.43 \pm 22.86$ | $254.76 \pm 169.97$ | $119.60 \pm 102.96$ | $209.50 \pm 153.83$ | $301.80 \pm 173.12$ | $570.31 \pm 231.27$ |
| Average | $241.34 \pm 181.00$ | $298.47 \pm 177.91$ | $275.22 \pm 190.57$ | $287.36 \pm 177.74$ | $250.58 \pm 189.78$ | $257.59 \pm 178.88$ | $566.59 \pm 240.22$ | $613.86 \pm 223.58$ |

Table 11: **Mean and standard deviation of token lengths** for correct and incorrect responses across four LLMs, broken down by instruction type. The consistent length difference between correct and incorrect responses indicates that length bias is prevalent in naturally generated responses, potentially affecting uncertainty estimation.

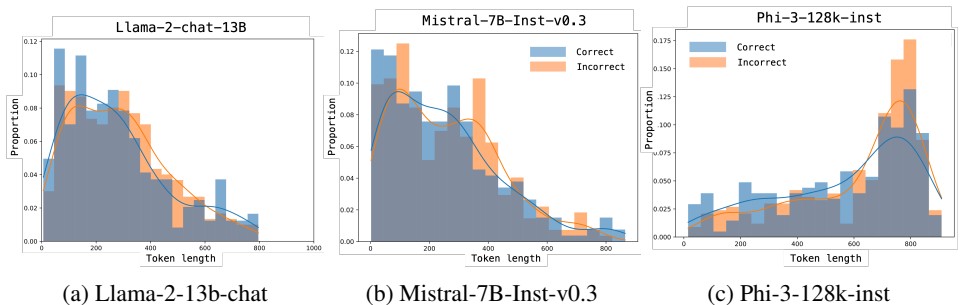

(a) Llama-2-13b-chat      (b) Mistral-7B-Inst-v0.3      (c) Phi-3-128k-inst

Figure 10: **Token length distributions** for LLaMA-2-chat-13B, Mistral-7B-Inst-v0.3, and Phi-3-128k-inst. The distributions are normalized by the total number of responses in each class. The distributions show that incorrect responses tend to be longer than correct ones, a pattern observed in existing datasets like IFEval, where naturally generated responses are used. This length signal highlights the potential bias in uncertainty estimation methods that are sensitive to token length.

