# OpenReview forum: "Do LLMs estimate uncertainty well in instruction-following?"
_ICLR.cc/2025/Conference — ICLR 2025 Poster_

### Official Review · Reviewer_8MWv · 2024-10-28

**Soundness:** 4
**Presentation:** 3
**Contribution:** 3
**Rating:** 8
**Confidence:** 3

**Summary:**

This paper studies uncertainty in instruction following, an important topic for interpretability and trustworthiness. It reveals the discrepancy between accuracy and uncertainty estimation, and among different uncertainty estimation methods themselves. It also accentuates the entanglement problem between instruction following and task completion.

Then the paper proposed a new benchmark separating control and realistic inferences, and shared more findings. IMO the most crucial one is that probe-based method provides the best estimation.

**Strengths:**

This paper reveals the uncertainty challenges faced by today's LLMs and presents new methods to better gauge LLM ability to estimate its own uncertainty. I think this is a timely work raising awareness to the community.

I'm not surprised by the top performance of probe-based method because internal states do not lie. Unclaimed by the authors, but the paper hints at open models' advantages to critical tasks, because its white box properties welcome scrutiny and checking by anyone. IMO internal states should be accessible by read APIs.

**Weaknesses:**

The poor uncertainty estimation by modern LLM is also not surprising because of its strong alliance with MLE methodology instead of MAP. Therefore, besides future directions outlined by authors, I strongly suggest trying more inference methods other than greedy, e.g. setting temperature, beam search, etc. These are also tunable parameters to closed-source models. This way you can measure closed models although the insights will be limited.

**Questions:**

Besides trying more inference method, please also consider impact of regularization such as dropout rate if you plan to do any pre- or post-training works. This reveals another problem which is training details are not sufficiently exposed even for open LLMs.

---

> ### Author Response · Authors · 2024-11-17
>
> Thank you for your thoughtful review and for recognizing the importance and timeliness of our work in addressing the uncertainty estimation challenges faced by current LLMs. We deeply appreciate your acknowledgment of our contributions to the community and your recognition of the potential in leveraging internal states to enhance uncertainty estimation in LLMs.
>
> ---
> ### W. Suggestion on future work
> > *The poor uncertainty estimation by modern LLM is also not surprising because of its strong alliance with MLE methodology instead of MAP. Therefore, besides future directions outlined by authors, I strongly suggest trying more inference methods other than greedy, e.g. setting temperature, beam search, etc. These are also tunable parameters to closed-source models. This way you can measure closed models although the insights will be limited.*
> >
>
> Thank you for this insightful suggestion. Exploring different sampling strategies at inference time, such as temperature adjustment and beam search, could indeed reveal further findings in LLMs’ uncertainty estimation abilities. While our current work focuses on greedy sampling for consistency across evaluations, extending the evaluation to include various sampling methods presents an exciting direction for future work, especially for closed-source models.
>
> ### Q. Suggestion on future work
> > *Besides trying more inference method, please also consider impact of regularization such as dropout rate if you plan to do any pre- or post-training works. This reveals another problem which is training details are not sufficiently exposed even for open LLMs.*
> >
>
> The impact of regularization, such as dropout rates, is another intriguing direction to consider. We agree that training details, including regularization techniques, can significantly influence LLMs’ uncertainty estimation capabilities. Thank you for highlighting these important points, which could guide future work.
>
> ---
>
> Thank you once again for your thoughtful feedback and for suggesting valuable directions to expand our work. We greatly appreciate your recognition of the significance of our study and the potential it holds for advancing uncertainty estimation in LLMs.

---

### Official Review · Reviewer_N6r8 · 2024-11-03

**Soundness:** 1
**Presentation:** 1
**Contribution:** 1
**Rating:** 1
**Confidence:** 3

**Summary:**

This paper evaluates the uncertainty estimation abilities of LLMs in instruction-following tasks. It introduces a controlled evaluation setup with two benchmark versions to isolate and compare methods under various conditions. Findings reveal that existing uncertainty methods struggle, especially with subtle errors, and that internal model states provide some improvement but remain inadequate in complex scenarios.

**Strengths:**

The study provides crucial insights into LLMs's uncertainty estimation in instruction-following problems.

**Weaknesses:**

1. The paper relies on a single IFEval for its initial evaluation. It would be beneficial to include additional datasets to validate the findings across different contexts and domains.
2. The paper raises many novel concepts and findings, but it does not seem to provide much direct help in enhancing instruction-following capabilities.
3. The article did not use LLMs larger than 13B in its tests, so the conclusions may not be sufficient.

**Questions:**

The study provides valuable insights into the limitations of LLMs in estimating uncertainty during instruction-following tasks. Given these findings, could you elaborate on specific strategies or methodologies that could be implemented to enhance the instruction-following accuracy of LLMs?

---

> ### Author Response · Authors · 2024-11-17
>
> Thank you for your review and for acknowledging that our work provides crucial insights into the LLMs’ uncertainty estimation in instruction-following tasks.
>
> ---
> ### W1. Dataset scope
> > *The paper relies on a single IFEval dataset for its initial evaluation. It would be beneficial to include additional datasets to validate the findings across different contexts and domains.*
> >
>
> Thank you for your comments. Please refer to the general response section for a more detailed explanation, where we address these aspects and discuss our rationale for the current benchmark design and its applicability to future datasets.
>
> ### W2 & Q
> > *The paper raises many novel concepts and findings, but it does not seem to provide much direct help in enhancing instruction-following capabilities.*
> >
>
> First of all, we would like to clarify that our focus is specifically on the “uncertainty estimation” capabilities of LLMs within instruction-following tasks, rather than directly improving instruction-following performance itself. While the two abilities are related, they address different aspects of model behavior. For instance, if the instruction is, “Please do not use punctuation in your response,” a model may struggle to comply with this if it conflicts with what it has learned in training — it is very unlikely to not include punctuation in training dataset. However, the model could still recognize and assess whether it has correctly followed the instruction, offering an uncertainty estimate on its compliance. In this case, the model demonstrates **low instruction-following ability** but **high uncertainty estimation ability.**
>
> To highlight our contribution once again in the uncertainty estimation side, our findings shed light on the strengths and limitations of widely used uncertainty estimation methods, potentially paving the way for improvements in how LLMs gauge their own uncertainty. For example, discovering that all methods struggle with subtle instruction-following errors highlights inherent limitations in LLMs. We hope this work will inspire further research to enhance LLMs’ ability to estimate uncertainty, ultimately leading to more reliable AI systems.
>
> ### W3. Concerns on models
> > *The article did not use LLMs larger than 13B in its tests, so the conclusions may not be sufficient.*
> >
>
> Due to computational constraints, we were unable to test models larger than 13B. We would like to share that there are many previous studies on trustworthy LLMs also focus on models up to 7B or 13B due to similar limitations [1, 2]. Instead, we included a variety of LLMs (e.g., Mistral, Phi, LLaMA) to capture a range of model architectures and configurations. We hope our benchmark dataset will facilitate further research, including evaluations of larger models in future studies.
>
> [1] Liu, Linyu, et al. "Uncertainty Estimation and Quantification for LLMs: A Simple Supervised Approach." *arXiv preprint arXiv:2404.15993* (2024).
>
> [2] Li, Kenneth, et al. "Inference-time intervention: Eliciting truthful answers from a language model." *Advances in Neural Information Processing Systems* 36 (2024).
>
> ---
>
> Thank you for your review and for acknowledging the insights our work provides into the uncertainty estimation capabilities of LLMs in instruction-following tasks. We hope our detailed responses clarify the focus and contributions of our study.

---

### Official Review · Reviewer_9E6i · 2024-11-03

**Soundness:** 3
**Presentation:** 3
**Contribution:** 3
**Rating:** 8
**Confidence:** 3

**Summary:**

This paper presents a first of its kind systematic evaluation of uncertainty estimation methods in instruction-following tasks, identifies key challenges in existing datasets and introduce two new benchmark datasets created to address those challenges. The results indicate that there is still a gap in the uncertainty estimation performance of common LLMs, especially when the errors made by the model are subtle. On top of this, internal model states, particularly the middle hidden layers, show an improvement but still remain unsatisfactory in complex scenarios. The main contributions and the motivation are presented clearly, with experiments performed on multiple LLMs and datasets.

**Strengths:**

- the paper introduces new benchmark datasets (Controlled and Realistic versions) that isolate factors influencing uncertainty estimation in instruction-following, filling an existing research gap.
- the methodologies employed are rigorous, with comprehensive experimental setups involving multiple LLMs and uncertainty estimation techniques.
- the writing is clear, and the results are presented in a way that is easy to follow, supported by well-designed figures and tables.
- the findings offer valuable insights for advancing the field, particularly the observation that models’ internal states can be leveraged for improved uncertainty estimation.

**Weaknesses:**

- the analysis might benefit from extending the scope of instruction types and domains included in the benchmark to cover more diverse real-world tasks
- while the paper identifies the use of internal states for uncertainty estimation as promising, it falls short in exploring more sophisticated methods that could leverage this information in nuanced tasks
- the use of GPT-4 for task quality assessment introduces a potential risk of pre-training overlap affecting the evaluation, though this is acknowledged as a limitation

**Questions:**

1. Can the authors elaborate on how training procedures like RLHF can affect the models' performance in uncertainty estimation, especially in Controlled-Hard tasks?
2. Are there potential methods beyond linear probing that could better utilize internal model states for uncertainty estimation in nuanced tasks?
3. Would expanding the evaluation to include LLM agents provide more comprehensive insights?

---

> ### Author Response · Authors · 2024-11-17
>
> Thank you for the positive and constructive feedback. We deeply appreciate your recognition of our contributions, particularly the development of new benchmark datasets that address critical gaps in uncertainty estimation for instruction-following tasks. We are pleased that you found the methodologies rigorous, the experimental setups comprehensive, and the writing and presentation clear. Your acknowledgment of our findings—especially the potential of leveraging models’ internal states for improved uncertainty estimation—motivates us to continue advancing this area. Thank you for highlighting these key aspects of our study.
>
> ---
>
> ### W1. Dataset scope
> > *the analysis might benefit from extending the scope of instruction types and domains included in the benchmark to cover more diverse real-world tasks*
> >
>
> Thank you for your thoughtful comments. Please refer to the general response section for a more detailed explanation, where we address these aspects and discuss our rationale for the current benchmark design and its applicability to future datasets.
>
> ### W3. GPT-4 evaluator
> > *the use of GPT-4 for task quality assessment introduces a potential risk of pre-training overlap affecting the evaluation, though this is acknowledged as a limitation*
> >
>
> Thank you for highlighting this point. The potential bias from using GPT-4 is indeed a valid concern. To address this limitation, we plan to investigate multiple large models as evaluators in future work, which would reduce the risk of bias that may arise from relying on a single model’s perspective. You can refer to Zeng et al. (2023) [1] for a detailed evaluation of the reliability of the GPT-4 evaluator.
> [1] Zeng, Zhiyuan, et al. "Evaluating large language models at evaluating instruction following." *arXiv preprint arXiv:2310.07641* (2023).
>
> ### Q1. Suggestion on future work
> > *Can the authors elaborate on how training procedures like RLHF can affect the models' performance in uncertainty estimation, especially in Controlled-Hard tasks?*
> >
>
> This is a thought-provoking question. Comparing models before and after reinforcement learning from human feedback (RLHF), specifically comparing models such as LLaMA2-7B and LLaMA2-chat-7B, seems very interesting and worth exploring. This will help clarify how RLHF might impact uncertainty estimation, especially under challenging Controlled-Hard settings, by influencing model confidence calibration and response tendencies. Thank you again for your suggestion.
>
> ### Q2. Suggestion on future work
> > *Are there potential methods beyond linear probing that could better utilize internal model states for uncertainty estimation in nuanced tasks?*
> >
>
> This is an interesting suggestion. We agree that more complex models could potentially capture nonlinear patterns in the representation space, providing a richer uncertainty signal. However, a critical consideration here is the need for a sufficiently large number of high-quality, objectively verifiable instruction-following datasets to avoid overfitting. We hope that as these datasets become available, future work can explore such models more robustly.
>
> ### Q3. Suggestion on future work
> > *Would expanding the evaluation to include LLM agents provide more comprehensive insights?*
> >
>
> This is another valuable idea. If we understand you correctly, evaluating LLM agents specifically fine-tuned or in-context trained for instruction-following tasks in particular domains would allow us to investigate whether domain-specific training impacts uncertainty estimation. Such training might improve uncertainty estimation ability, or lead to overconfidence in uncertainty estimation, potentially affecting reliability in nuanced tasks. We hope to explore this direction in future work and thank you for the suggestion.
>
> ---
>
> Thank you again for your thoughtful and constructive feedback. Your detailed suggestions and acknowledgment of our contributions reinforce the significance of our work and provide valuable guidance for future research directions.

---

> > ### Comment · Reviewer_8MWv · 2024-11-26
> >
> > Thank you for the response.

---

### Official Review · Reviewer_zm1y · 2024-11-04

**Soundness:** 3
**Presentation:** 2
**Contribution:** 3
**Rating:** 6
**Confidence:** 2

**Summary:**

The paper evaluates how well LLMs can estimate uncertainty when tasked with instruction following, given that existing research largely focuses on uncertainty in factuality. The authors systematically evaluate the uncertainty estimation methods using the IFEval dataset and further introduce a new benchmark to disentangle the multiple factors. Experimental results highlight the potential of self-evaluation and probing methods, though they still fall short under difficult conditions, underscoring the need for further research on uncertainty estimation.

**Strengths:**

1. The authors provide a systematic evaluation, identifying multiple factors that affect uncertainty estimation and are entangled within naturally generated responses, which helps isolate the influence of various factors and allows for a more accurate assessment of estimation methods' capabilities.
2. The study offers thorough comparisons between different uncertainty estimation methods, various LLMs, and distinct datasets.

**Weaknesses:**

1. The Probe method is mentioned in Table 4 and highlighted as the best-performing method. However, it is not introduced or discussed in Section 3, leading to some confusion when comparing results in Section 3.2.1. A clearer explanation of this method earlier in the paper would improve the comprehensibility of the findings. Or at least keep the scope of methods consistent when discussing "best-performing".
2. As noted by the authors in their limitations section, the types of instructions included in the benchmark may not comprehensively reflect real-world tasks. Additionally, the dataset size is relatively small, potentially impacting the reliability of the benchmark.

**Questions:**

1. Could you please provide more details in benchmark data generation process? For example, what's the prompt for generating correct response in controlled version, and natural response in realistic version? What are the domains for the arguments (e.g., Type, Kwargs) in the prompts? Are these parameters inherited from IFEval?
2. It seems that all the token length distributions are not normalized by the total number of total number of responses in each class (Figure 2, 3, and 10). Or to say, the total numbers of correct and incorrect responses are not the same, which means you may not draw a conclusion that incorrect responses tend to be longer than correct ones when only seeing that the orange curve is significantly higher than the blue curve on larger tokens. Normalizing the distribution would be better, though it may not affect the conclusions that token lengths introduce biases.

---

> ### Author Response · Authors · 2024-11-17
>
> Thank you for your constructive feedback and for acknowledging our systematic approach in devising a controlled evaluation framework. By isolating multiple factors that impact uncertainty estimation in naturally generated responses, we aim to provide a clearer and more fine-grained assessment of each method’s performance.
>
> ---
> ### W1. Clarification
> >*The Probe method is mentioned in Table 4 and highlighted as the best-performing method. However, it is not introduced or discussed in Section 3, leading to some confusion when comparing results in Section 3.2.1. A clearer explanation of this method earlier in the paper would improve the comprehensibility of the findings. Or at least keep the scope of methods consistent when discussing "best-performing"*
>
> Thank you for this suggestion. We agree that introducing the Probe method earlier in the paper would clarify the findings. In our revision, we will integrate Section 4 into Section 3 to provide a consistent explanation of all methods, including Probe, for improved clarity in comparisons.
>
> ### W2. About dataset scope
> > *As noted by the authors in their limitations section, the types of instructions included in the benchmark may not comprehensively reflect real-world tasks. Additionally, the dataset size is relatively small, potentially impacting the reliability of the benchmark.*
> >
>
> Thank you for your thoughtful feedback. For a more detailed explanation, please refer to the general response section, where we address these aspects and discuss our rationale for the current benchmark design and its applicability to future datasets.
>
> ### Q1. Details in benchmark data generation process
> > *Could you please provide more details in the benchmark data generation process? For example, what’s the prompt for generating a correct response in the controlled version and natural responses in the realistic version? What are the domains for the arguments (e.g., Type, Kwargs) in the prompts? Are these parameters inherited from IFEval?*
> >
>
> We really appreciate your question in the detailed benchmark generation process. Yes, the argument structures in our prompts (such as Type and Kwargs) are inherited from the IFEval dataset. Type specifies the instruction type (e.g., ‘keywords:forbidden_words’, etc.), while Kwargs details task-specific parameters (e.g.,{”forbidden_words”:[”night”, “happy”, …]}). Full versions of these prompts are available in the Appendix. We will clarify this in the revision.
>
> ### Q2. Normalization of token length
> > *It seems that all the token length distributions are not normalized by the total number of responses in each class (Figure 2, 3, and 10). This may mean that incorrect responses appear to be longer simply because of different class sizes. Normalizing the distributions would help, even if it may not change the conclusions about length biases.*
> >
>
> Thank you for noting this. We agree that normalizing token length distributions would clarify our conclusions. We have since normalized these distributions and found that, as expected, the conclusions remain consistent. We will include these normalized plots in the revised submission.
>
>
>
> ---
> Thank you again for recognizing the importance of our contributions and for your thoughtful suggestions to improve the paper.

---

> > ### Comment · Reviewer_zm1y · 2024-11-25
> >
> > I sincerely thank the authors for their clarifications. However, I am maintaining my current rating as some concerns are being addressed in a revised version not yet submitted. Since many of these issues do not require further experiments, I expect the authors to resolve them more clearly.
> >
> > Specifically, addressing W1 will significantly improve the clarity, and correcting all the figures mentioned in Q2 will improve the soundness. Regarding Q1, it will be better to still introduce the arguments inherited from the IFEval dataset in the Appendix. The authors claimed that "full versions of these prompts are available in the Appendix", but I could only find the prompts for generating subtle off-target responses and totally wrong responses. Could the authors please point me to the prompts for "generating a correct response in the controlled version and natural responses in the realistic version"? For W2, I sincerely thank the authors for their clarification in the General Response, and I think adding them into the revised version will further improve the manuscript.

---

> > > ### Author Response · Authors · 2024-11-25
> > >
> > > We sincerely thank you for your thoughtful feedback and detailed suggestions. We have now uploaded the revised version of the manuscript, incorporating changes in response to your comments.
> > >
> > > ---
> > >
> > > ### W1. Integrated Sections
> > >
> > > To improve clarity and ensure consistent comparisons, we have integrated Section 4 into Section 3, providing a unified explanation of all methods, including Probe.
> > >
> > > ### W2. Data Scope
> > >
> > > We have added the explanation regarding the data scope to Section A.4 of the Appendix, further enhancing the manuscript’s completeness.
> > >
> > > ### Q1. Prompts for Generating Benchmark Data
> > >
> > > For generating correct responses, we used plain prompts from the IFEval data. However, for improved clarity, we have now explicitly added the prompts for generating correct responses in both the Controlled and Realistic versions of the benchmark dataset in Section A.3 of the Appendix. Additionally, we have included examples from the IFEval dataset in Section A.5 (IFEval Data Examples) for better context and transparency.
> > >
> > >
> > > ### Q2. Normalized Figures
> > >
> > > Figures 2, 3, and 10 have been updated after normalizing the token length distributions by the total number of responses in each class. These updates directly address the soundness concerns raised in Q2.
> > >
> > > ---
> > >
> > > We hope these revisions address your concerns and significantly enhance the quality of the manuscript. Thank you once again for your invaluable feedback.

---

> > > > ### Comment · Reviewer_zm1y · 2024-11-26
> > > >
> > > > Thank the authors for their responses, which enhance the clarity and strength of their work. I am increasing my rating.

---

> > > > > ### Author Response · Authors · 2024-11-26
> > > > >
> > > > > Thank you for your thoughtful review and for taking the time to consider our responses. Your feedback has been invaluable in helping us refine the manuscript, and we are grateful for your support.

---

### Author Response · Authors · 2024-11-17
**General response**

# General response
Thank you for all reviewers thoughtful and constructive feedback on our work. We greatly appreciate reviewers recognition of our contributions, particularly our systematic evaluation of uncertainty estimation in instruction-following tasks, which disentangles multiple factors within naturally generated responses to enable more accurate assessments. Many of reviewers highlighted the novelty and significance of our benchmark datasets—Controlled and Realistic—which address key challenges and fill an important research gap. We are pleased that our rigorous methodology, comparing multiple uncertainty estimation methods across diverse LLMs and datasets, and the valuable insights provided by our findings, were seen as strengths. We also appreciate your positive remarks on the clarity and accessibility of our writing, supported by well-designed figures and tables. Reviewers feedback inspires us to refine this work further and continue advancing research in this critical area.


---

### Q. Dataset scope

>*Reviewer 9E6i: "The analysis might benefit from extending the scope of instruction types and domains included in the benchmark to cover more diverse real-world tasks."*

>*Reviewer zm1y: "As noted by the authors in their limitations section, the types of instructions included in the benchmark may not comprehensively reflect real-world tasks. Additionally, the dataset size is relatively small, potentially impacting the reliability of the benchmark."*

>*Reviewer N6r8: "The paper relies on a single IFEval for its initial evaluation. It would be beneficial to include additional datasets to validate the findings across different contexts and domains."*

Thank you for your concern about datasets. We do agree it would be beneficial to include real-world datasets to validate the findings. However, here we would like to emphasize why we choose IFEval as our primary dataset instead of using real-world dataset with different contexts and domains.

**First, we select IFEval to focus on our scope which is ‘single, simple, and non-ambiguous instructions’.** Real-world datasets often involve complex, ambiguous, or multi-instruction prompts, which can conflate multiple factors affecting uncertainty. For this first systematic evaluation, we chose to focus on single, simple, and verifiable instructions to ensure clarity and isolate the uncertainty estimation process. The IFEval dataset is well-suited for this purpose, as it provides 25 distinct types of simple and clear instructions that align with our goal of establishing a robust baseline.

**Second, we want to avoid evaluator-induced uncertainties**. Most real-world tasks and benchmark datasets rely on LLM-based evaluators to determine whether a response follows an instruction. However, LLM-based evaluators may introduce their own uncertainties or make errors in assessing success or failure, which could obscure the true uncertainty estimation capabilities of the tested models. The IFEval dataset avoids this issue by including instructions with deterministic evaluation programs that objectively verify compliance. For instance, an instruction like “please do not include keywords: …” can be automatically validated using a simple program to check for the presence of those keywords. This feature eliminates ambiguity in evaluation and allows us to directly focus on LLMs’ ability to estimate uncertainty.

Our main contribution is the careful design of benchmark data specifically tailored to ‘uncertainty estimation’ in instruction-following contexts. We believe that those underlying considerations (e.g., length bias, task quality entanglement, and varying levels of model difficulty) presented in our study can extend to future instruction-following datasets. **While IFEval serves as an ideal starting point for this research, we hope our framework inspires future efforts to tackle uncertainty estimation in more complex, real-world tasks.**

---

### Meta-Review · Area_Chair_yj6b · 2024-12-19

**Metareview:**

The paper introduces a controlled benchmark to assess how LLMs estimate their uncertainty when following instructions. All of the reviewers (except one) agree that the paper is above the bar for publication at ICLR. The one dissenting reviewer did not back up their score with constructive feedback, and their reviewed weaknesses appear to not be in good faith.

The reviewers appreciated the novel and sound methodology for evaluation, and the study of a well motivated problem. The paper could also spur interesting follow-up works (e.g. evaluating the effect of RLHF interventions, devising ways to improve uncertainty calibration during instruction following etc.).

There were good suggestions for improving the clarity and exposition of the paper, that the authors acknowledged in their rebuttal and promised a revision.

**Additional Comments On Reviewer Discussion:**

The reviewers asked for additional details around the prompts used for controlled generation that the authors provided.

There are two main weaknesses that were not fully addressed during the rebuttal period:
1. Expanding the scope of instruction types beyond IFEval. The authors reiterated the benefit of limiting the scope to IFEval, and punted the concern (shared by all the reviewers) of broadening the scope of instructions to future work.
2. Using GPT4o as evaluator. The reviewers pointed out the potential for bias that the authors acknowledged. They pointed to other papers that use GPT4 as evaluators, and promise as future work to use a variety of LLMs as evaluators, as well as systematically study whether there is potential bias in this application of them.

---

### Decision · Program_Chairs · 2025-01-22

Accept (Poster)